# Navigation-Guided Sparse Scene Representation for End-to-End Autonomous Driving

**Peidong Li, Dixiao Cui**
Zhijia Technology, Suzhou, China
{lipeidong,cuidixiao}@smartxtruck.com

## Abstract

End-to-End Autonomous Driving (E2EAD) methods typically rely on supervised perception tasks to extract explicit scene information (e.g., objects, maps). This reliance necessitates expensive annotations and constrains deployment and data scalability in real-time applications. In this paper, we introduce **SSR**, a novel framework that utilizes only 16 navigation-guided tokens as **S**parse **S**cene **R**epresentation, efficiently extracting crucial scene information for E2EAD. Our method eliminates the need for human-designed supervised sub-tasks, allowing computational resources to concentrate on essential elements directly related to navigation intent. We further introduce a temporal enhancement module, aligning predicted future scenes with actual future scenes through self-supervision. SSR achieves a 27.2% relative reduction in L2 error and a 51.6% decrease in collision rate to UniAD in nuScenes, with a 10.9× faster inference speed and 13× faster training time. Moreover, SSR outperforms VAD-Base with a 48.6-point improvement on driving score in CARLA's Town05 Long benchmark. This framework represents a significant leap in real-time autonomous driving systems and paves the way for future scalable deployment. Code is available at https://github.com/PeidongLi/SSR.

## 1 Introduction

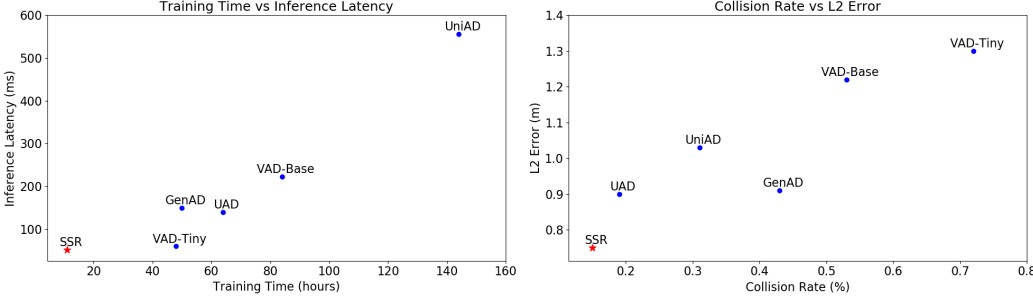

Figure 1: **Performance Comparison of Various Methods in Speed and Accuracy on nuScenes.**

Vision-based E2EAD (Hu et al., 2023; Jiang et al., 2023; Sima et al., 2023; Zheng et al., 2024b; Sun et al., 2024; Weng et al., 2024; Li et al., 2024b; Guo et al., 2024) has gained significant attention in recent research as a cost-effective alternative for autonomous driving systems. Traditional architectures typically consist of separate perception and planning modules. While most perception modules are handled by neural networks (NN), planning modules often rely on rule-based pipelines. This separation can lead to information loss during the transfer between modules, resulting in suboptimal performance. E2EAD addresses this by using entire neural networks to predict planning trajectories from images, thereby minimizing information loss and improving overall performance.

However, most current E2EAD approaches build upon complex perception frameworks, often incorporating additional NN-based planning modules. These approaches typically inherit tasks such as

object detection (Li et al., 2022b; Philion & Fidler, 2020; Li et al., 2024a), mapping (Li et al., 2022a; Liao et al., 2022), and occupancy prediction (Sima et al., 2023; Huang et al., 2023), resulting in large and computationally intensive neural networks. Despite their integration, these models still maintain modular framework design, requiring independent sub-tasks' supervision. As a result, they remain annotation-intensive, suffer from scalability issues, and are inefficient for real-time deployment.

While many E2EAD methods continue to mimic the paradigm established by prior BEV perception works, they often overlook a critical question: *Do E2EAD systems still require such extensive perception tasks?* In traditional AD systems, the perception module has to extract all elements for the planning module, as there is no back-propagation from planning to perception. Existing E2EAD methods largely ignore the advanced planning-oriented insight of E2E paradigms, instead retaining a cascade structure rooted in traditional AD. In contrast, we seek for a more targeted approach that directly identifies driving-relevant elements. This raises a key question: ***How can we efficiently identify and focus on the crucial parts of the scene without auxiliary perception supervision?***

To address this, we introduce SSR, a novel framework that leverages navigation-guided **S**parse **S**cene **R**epresentation, learning from temporal context in self-supervision rather than explicit perception supervision. Inspired by how human drivers selectively focus on scene elements based on navigation cues, we find that only a minimal set of tokens from dense BEV features is necessary for effective scene representation in autonomous driving. Since E2EAD methods do not rely on high-definition maps as input, a high-level command (e.g., "*turn left*", "*turn right*", "*go straight*" following common practices in Hu et al. (2023); Jiang et al. (2023)) is required for navigation. Our method, therefore, extracts scene queries guided by the navigation commands, akin to human attention mechanisms.

As illustrated in Fig. 2(a), existing methods typically extract all perception elements following previous BEV perception paradigms. These methods rely on Transformer (Vaswani et al., 2017) to identify relevant ones in the additional planning stage. In contrast, as shown in Fig. 2(b), SSR directly extracts only the essential perception elements in the guidance of navigation commands, thereby minimizing redundancy. Our approach takes full advantage of the end-to-end framework, breaking the modular cascade architecture in a **Navigation-Guided Perception** manner. While prior works (Sun et al., 2024; Zhang et al., 2024) attempt to reduce computation by skipping BEV feature construction, they still depend on hundreds of task-specific queries. However, our method drastically reduces computational overhead by using just 16 tokens guided by navigation commands.

Additionally, SSR capitalizes on temporal context to circumvent the need for perception tasks supervision. We hypothesize that if a predicted action aligns with the actual action, the resulting scene should match the real future scene. Specifically, we predict future BEV features, which are then self-supervised by the actual future BEV features. This future feature predictor, which takes current BEV features and the planning trajectory as input to predict future BEV features, offers an alternative for supervising both scene representation and planning trajectories without auxiliary annotations.

By leveraging navigation-guided perception paradigm and temporal self-supervision, SSR provides an effective and efficient solution for real-time autonomous driving. As illustrated in Fig. 1, SSR delivers state-of-the-art performance on the nuScenes (Caesar et al., 2020) dataset, with minimal computational overhead. Specifically, our method decreases average L2 error by 0.28 meters (a 27.2% relative improvement) and reduces the average collision rate by 51.6% relatively compared to UniAD (Hu et al., 2023), even without any annotations. Meanwhile, SSR also achieves superior performance on CARLA's (Dosovitskiy et al., 2017) Town05 Long benchmark. Remarkably, our method reduces training time to 1/13th of that required by UniAD and is 10.9× faster during inference. Therefore, SSR has the potential to manage large-scale data in real-time applications.

Our contributions are summarized as follow:

- We introduce a human-inspired E2EAD framework that utilizes learned sparse query representations guided by navigation commands, significantly reduces computational costs by adaptively focusing on essential parts of scenes.

- We highlight the critical role of temporal context in autonomous driving by introducing a future feature predictor for self-supervision on dynamic scene changes, eliminating the need for costly perception tasks supervision.

- Our framework achieves state-of-the-art performance on both open-loop and closed-loop experiments, establishing a new benchmark for real-time E2EAD.

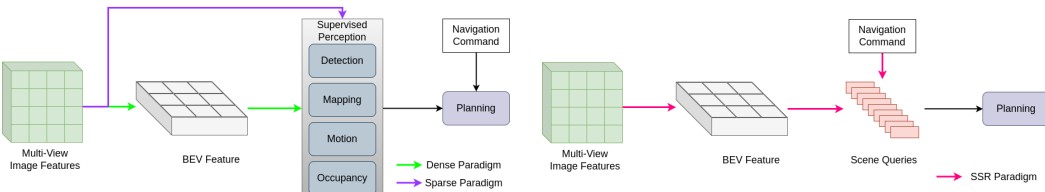

(a) Task-Specific Supervised Paradigm          (b) Adaptive Unsupervised Paradigm

Figure 2: **Comparison of Various End-to-End Paradigms.** Compared to previous task-specific supervised paradigms, our adaptive unsupervised approach takes full advantage of end-to-end framework by utilizing navigation-guided perception, without the need to differentiate between sub-tasks.

## 2    RELATED WORK

### 2.1    VISION-BASED END-TO-END AUTONOMOUS DRIVING

Research on End-to-End autonomous driving dates back to 1988 with ALVINN (Pomerleau, 1988), which used a simple neural network to generate steering outputs. NVIDIA developed a prototype E2E system (Bojarski et al., 2016) based on convolutional neural networks (CNN), bypassing manual decomposition. The recent resurgence in vision-based E2EAD has been driven by rapid advancements in BEV perception (Li et al., 2022b; Liao et al., 2022; Liu et al., 2022; Huang et al., 2023) and modern architectures like Transformer (Vaswani et al., 2017).

ST-P3 (Hu et al., 2022) introduced improvements in perception, prediction, and planning modules for enhanced spatial-temporal feature learning, integrating auxiliary tasks such as depth estimation and BEV segmentation. UniAD (Hu et al., 2023) built on previous BEV perception works to create a cascade framework with a variety of auxiliary tasks, including detection, tracking, mapping, occupancy, and motion estimation. VAD (Jiang et al., 2023) sought to streamline scene representation by vectorizing it, reducing the tracking and occupancy tasks seen in UniAD. GenAD (Zheng et al., 2024b) explored the use of generative models for trajectory generation, jointly optimizing motion and planning heads based on VAD. PARA-Drive (Weng et al., 2024) further examined the relationship between auxiliary tasks, reorganizing them to run in parallel while deactivating them during inference. In contrast, our approach eliminates all perception tasks, achieving remarkable performance in both accuracy and efficiency.

### 2.2    SCENE REPRESENTATION IN AUTONOMOUS DRIVING

Most prior works in autonomous driving (Hu et al., 2022; 2023; Jiang et al., 2023; Zheng et al., 2024b) have inherited approaches from perception tasks, such as Li et al. (2022b), which leverages dense BEV features as the primary scene representation. In these frameworks, task-specific queries (e.g., for detection and mapping) are used to extract information from the BEV features under manual labels' supervision. While these approaches benefit from rich scene information, they also introduce significant model complexity, hindering real-time application, particularly in occupancy-based representations (Sima et al., 2023; Zheng et al., 2024a).

Following the trend of sparse paradigms in BEV detection (Lin et al., 2022; Liu et al., 2023), recent sparse E2EAD approaches (Sun et al., 2024; Zhang et al., 2024) directly utilize task-specific queries to interact with image features. These methods attempt to bypass BEV feature generation altogether by directly interacting with image features through task-specific queries. However, despite the reduction in BEV processing, these models still rely on hundreds of queries, which diminish the promised simplicity and efficiency of the end-to-end paradigm. LAW (Li et al., 2024b) proposed the use of view latent queries to represent each camera image with a single query. However, this approach compromises information fidelity, leading to suboptimal performance. UAD (Guo et al., 2024) attempted to divide the BEV feature into angular-wise sectors but still relied on open-set detector labels for supervision, maintaining the complexity of task-specific queries. In this work, we introduce SSR, a novel approach that represents the scene by a minimal set of adaptively learned queries, enhancing both efficiency and performance.

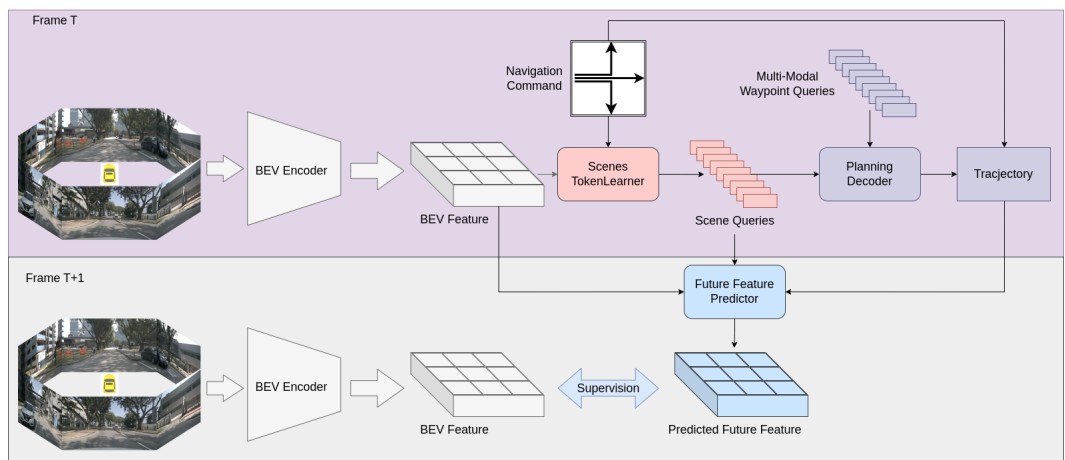

Figure 3: **Overview of SSR:** SSR consists of two parts: the purple part, which is used during both training and inference, and the gray part, which is only used during training. In the purple part, the dense BEV feature is first compressed by the Scenes TokenLearner into sparse queries, which are then used for planning via cross-attention. In the gray part, the predicted BEV feature is obtained from the Future Feature Predictor. The future BEV feature is then used to supervise the predicted BEV feature, enhancing both the scene representation and the planning decoder.

## 3 METHOD

### 3.1 OVERVIEW

**Problem Formulation:** At timestamp $t$, given surrounding N-views camera images $\mathbf{I}_t = [\mathbf{I}_t^i]_{i=1}^N$ and a high-level navigation command $cmd$, the vision-based E2EAD model aims to predict the planning trajectory $\mathbf{T}$, which consists of a set of points in BEV space.

**BEV Feature Construction:** As shown in Fig. 3, N-views camera images $\mathbf{I}_t$ are processed by a BEV encoder to generate the BEV feature. In the BEV encoder such as BEVFormer (Li et al., 2022b), $\mathbf{I}_t$ is first processed by an image backbone to obtain image features $\mathbf{F}_t = [\mathbf{F}_t^i]_{i=1}^N$. A BEV query $\mathbf{Q} \in \mathbb{R}^{H \times W \times C}$ is then used to query temporal information from the previous frame's BEV feature $\mathbf{B}_{t-1}$ and spatial information from $\mathbf{F}_t$ by cross-attention iteratively, resulting in the current BEV feature $\mathbf{B}_t \in \mathbb{R}^{H \times W \times C}$. Here, $H \times W$ represents the spatial dimensions of the BEV feature, and $C$ denotes the feature's channel dimension.

$$\mathbf{Q} = CrossAttention(\mathbf{Q}, \mathbf{B}_{t-1}, \mathbf{B}_{t-1}), \tag{1}$$

$$\mathbf{B}_t = CrossAttention(\mathbf{Q}, \mathbf{F}_t, \mathbf{F}_t). \tag{2}$$

The key component of our framework is the novel Scenes TokenLearner module to extract crucial scene information, introduced in Sec. 3.2. Unlike traditional methods that rely on dense BEV features or hundreds of queries, our approach uses a small set of tokens to effectively represent the scene. Leveraging these sparse scene tokens, we generate the planning trajectory, a process detailed in Sec. 3.3. Additionally, in Sec. 3.4, we introduce an augmented future feature predictor designed to further enhance the scene representation through self-supervised learning of BEV features.

### 3.2 NAVIGATION-GUIDED SCENES TOKEN LEARNER

BEV features are a popular scene representation as they contain rich perception information. However, this dense representation increases the inference time when searching for relevant perception elements. To address this, we introduce a sparse scene representation using adaptive spatial attention, significantly reducing computational load while maintaining high-fidelity scene understanding.

Specifically, we propose the Scenes TokenLearner (STL) module to extract scene queries $\mathbf{S}_t = [\mathbf{s}_i]_{i=1}^{N_s} \in \mathbb{R}^{N_s \times C}$ from the BEV feature, where $N_s$ is the number of scene queries, to efficiently

Figure 4: **Structure of Modules: Scenes TokenLearner and Future Feature Predcitor**.

represent the scene. The structure of Scenes TokenLearner is illustrated in Fig. 4. To better focus on scene information related to our navigation intent, we adopt a Squeeze-and-Excitation (SE) layer (Hu et al., 2018) to encode the navigation command *cmd* into the dense BEV feature, producing the navigation-aware BEV feature $\mathbf{B}_t^{navi}$:

$$\mathbf{B}_t^{navi} = SE(\mathbf{B}_t, cmd). \tag{3}$$

The navigation-aware BEV feature is then passed into the BEV TokenLearner (Ryoo et al., 2021) module $TL_{BEV}$ to adaptively focus on the most important information. Unlike previous applications of TokenLearner in image or video domains, we utilize it in BEV space to derive a sparse scene representation via spatial attention:

$$\mathbf{S}_t = TL_{BEV}(\mathbf{B}_t^{navi}). \tag{4}$$

For each scene query $\mathbf{s}_i$, we adopt a tokenizer function $M_i$ that maps $\mathbf{B}_t^{navi}$ into a token vector: $\mathbb{R}^{H \times W \times C} \to \mathbb{R}^C$. The tokenizer predicts spatial attention maps of shape $H \times W \times 1$, and the learned scene tokens are obtained through global average pooling:

$$\mathbf{s}_i = M_i(\mathbf{B}_t^{navi}) = \rho(\mathbf{B}_t^{navi} \odot \varpi_i(\mathbf{B}_t^{navi})), \tag{5}$$

where $\varpi(\cdot)$ is the spatial attention function and $\rho(\cdot)$ is the global average pooling function. The multi-layer self-attention (Vaswani et al., 2017) is applied to further enhance the scene queries:

$$\mathbf{S}_t = SelfAttention(\mathbf{S}_t). \tag{6}$$

### 3.3 PLANNING BASED ON SPARSE SCENE REPRESENTATION

Since $\mathbf{S}_t$ contains all relevant perception information, we use a set of way point queries $\mathbf{W}_t \in \mathbb{R}^{N_m \times N_t \times C}$ to extract multi-modal planning trajectories, where $N_t$ is the number of future timestamps and $N_m$ denotes the number of driving commands.

$$\mathbf{W}_t = CrossAttention(\mathbf{W}_t, \mathbf{S}_t, \mathbf{S}_t). \tag{7}$$

We then obtain the predicted trajectory from $\mathbf{W}_t$ using a multi-layer perceptron (MLP), and select output trajectory $\mathbf{T} \in \mathbb{R}^{N_t \times 2}$ based on the navigation command *cmd*:

$$\mathbf{T} = Select(MLP(\mathbf{W}_t), cmd). \tag{8}$$

The output trajectory is supervised by the ground truth (GT) trajectory $\mathbf{T}_{GT}$ using L1 loss, defined as the imitation loss $\mathcal{L}_{imi}$:

$$\mathcal{L}_{imi} = \|\mathbf{T}_{GT} - \mathbf{T}\|_1. \tag{9}$$

### 3.4 TEMPORAL ENHANCEMENT BY FUTURE FEATURE PREDICTOR

We prioritize temporal context to enhance scene representation by self-supervision instead of perception sub-tasks. The motivation behind this module is straight: if our predicted actions correspond to real actions, the predicted future scenes should closely resemble the actual future scenes.

As illustrated in Fig. 4, we introduce the Future Feature Predictor (FFP) to predict future BEV features. First, we use the output trajectory $\mathbf{T}$ to translate the current scene queries into the future

frame using a Motion aware Layer Normalization (MLN) (Wang et al., 2023) module. The MLN module helps current scene queries encode motion information, producing dreaming queries $\mathbf{D}_t$:

$$\mathbf{D}_t = MLN(\mathbf{S}_t, \mathbf{T}). \tag{10}$$

We then apply multi-layer self-attention on $\mathbf{D}_t$ to predict the future scene queries $\hat{\mathbf{S}}_{t+1}$:

$$\hat{\mathbf{S}}_{t+1} = SelfAttention(\mathbf{D}_t). \tag{11}$$

However, since the autonomous driving system may focus on different regions even in consecutive frames, we do not directly supervise the predicted scene queries $\hat{\mathbf{S}}_{t+1}$ with the future scene queries $\mathbf{S}_{t+1}$. Instead, we reconstruct the dense BEV feature $\hat{\mathbf{B}}_{t+1}$ using TokenFuser (Ryoo et al., 2021):

$$\hat{\mathbf{B}}_{t+1} = TokenFuser(\hat{\mathbf{S}}_{t+1}, \mathbf{B}_t) \tag{12}$$

$$= \psi(\mathbf{B}_t) \otimes \hat{\mathbf{S}}_{t+1}, \tag{13}$$

where $\psi(\cdot)$ is a simple MLP with the sigmoid function to remap the BEV feature $\mathbf{B}_t$ to a weight tensor: $\mathbb{R}^{H \times W \times C} \to \mathbb{R}^{H \times W \times N_s}$. After the multiplication $\otimes$ with $\hat{\mathbf{S}}_{t+1} \in \mathbb{R}^{N_s \times C}$, we obtain the predicted dense BEV feature $\hat{\mathbf{B}}_{t+1} \in \mathbb{R}^{H \times W \times C}$. This process aims to recover the BEV feature from the predicted scene queries for further self-supervision.

Unlike prior works (Zong et al., 2023; Zou et al., 2024) which utilize predicted BEV features for subsequent object- or pixel-level supervision, we supervise $\hat{\mathbf{B}}_{t+1}$ directly using an L2 loss with the real future BEV feature $\mathbf{B}_{t+1}$. This is defined as the BEV reconstruction loss $\mathcal{L}_{bev}$:

$$\mathcal{L}_{bev} = \|\hat{\mathbf{B}}_{t+1} - \mathbf{B}_{t+1}\|_2. \tag{14}$$

In summary, we apply imitation loss $\mathcal{L}_{imi}$ for the predicted trajectory, and BEV reconstruction loss $\mathcal{L}_{bev}$ for the predicted BEV feature. The total loss of SSR is:

$$\mathcal{L}_{total} = \mathcal{L}_{imi} + \mathcal{L}_{bev}. \tag{15}$$

## 4 EXPERIMENTS

### 4.1 DATASET AND METRIC

**Open-Loop** We evaluate the proposed SSR framework for autonomous driving using the widely adopted nuScenes dataset (Caesar et al., 2020), following prior works (Hu et al., 2023; Jiang et al., 2023). To assess planning performance, we use displacement error and collision rate (CR), as in previous studies. Displacement error is calculated by L2 error with respect to the GT trajectory, measuring the quality of predicted trajectory. Collision rate quantifies the percentage of collisions with other objects when following the predicted trajectory. All metrics are calculated in 3s future horizon with a 0.5s interval and evaluated at 1s, 2s and 3s.

We observe that VAD (Jiang et al., 2023) and UniAD (Hu et al., 2023) utilize different evaluation methods to calculate results across all predicted frames. VAD computes the average across all previous frames, while UniAD uses the latest result as well as the maximum value. Additionally, UniAD excludes pedestrians from the GT occupancy map, resulting in lower collision rates. We denote the VAD approach with the subscript AVG and the UniAD approach with MAX. For example, the L2 error at frame $t$ (maximum $3s/0.5s = 6$) is calculated as $L2^t_{AVG} = \frac{1}{t}\sum_{i=1}^{t} L2^i$ for VAD and $L2^t_{MAX} = L2^t$ for UniAD. We apply MAX metric by default but also calculate AVG metric for comparison with other methods in Tab. 1. In our MAX metric, pedestrians are included in the calculation of collision rate.

**Closed-Loop** We conduct closed-loop experiments using the CARLA simulator (Dosovitskiy et al., 2017), leveraging the widely adopted Town05 Long benchmark to evaluate performance. The training dataset consists of 189K frames collected by Roach (Zhang et al., 2021) at 2 Hz across 4 CARLA towns (Town01, Town03, Town04, and Town06), following previous works (Jia et al., 2023a;b; Wu et al., 2022). The training data has no overlap with Town05 Long benchmark.

We utilize the official metric provided by CARLA. The Route Completion (RC) is the percentage of the route completed by the autonomous agent. The Infraction Score (IS) quantifies the number of infractions made along the route, with pedestrians, vehicles, road layouts, and traffic signals. The Driving Score (DS) serves as the main metric, calculated as the product of RC and IS.

Table 1: **Comparison of state-of-the-art methods on the nuScenes dataset**. The ego status was not utilized in the planning module. ⋄: Lidar-based methods. ∗: Backbone with ResNet-101 (He et al., 2016), while others use ResNet-50 or similar. †: FPS measured on an NVIDIA A100 GPU, while others were tested on an NVIDIA RTX 3090. ‡: AVG metric protocal as same as VAD.

| Method | Auxiliary Task | L2 (m) ↓ | | | | Collision Rate (%) ↓ | | | | FPS |
|---|---|---|---|---|---|---|---|---|---|---|
| | | 1s | 2s | 3s | Avg. | 1s | 2s | 3s | Avg. | |
| NMP⋄ (Zeng et al., 2019) | Det & Motion | 0.53 | 1.25 | 2.67 | 1.48 | 0.04 | 0.12 | 0.87 | 0.34 | - |
| FF⋄ (Hu et al., 2021) | FreeSpace | 0.55 | 1.20 | 2.54 | 1.43 | 0.06 | 0.17 | 1.07 | 0.43 | - |
| EO⋄ (Khurana et al., 2022) | FreeSpace | 0.67 | 1.36 | 2.78 | 1.60 | 0.04 | 0.09 | 0.88 | 0.33 | - |
| ST-P3 (Hu et al., 2022) | Det & Map & Depth | 1.72 | 3.26 | 4.86 | 3.28 | 0.44 | 1.08 | 3.01 | 1.51 | 1.6 |
| UniAD∗ (Hu et al., 2023) | Det&Track&Map&Motion&Occ | 0.48 | 0.96 | 1.65 | 1.03 | 0.05 | 0.17 | 0.71 | 0.31 | 1.8† |
| OccNet∗ (Sima et al., 2023) | Det & Map & Occ | 1.29 | 2.13 | 2.99 | 2.14 | 0.21 | 0.59 | 1.37 | 0.72 | 2.6 |
| VAD-Base (Jiang et al., 2023) | Det & Map & Motion | 0.54 | 1.15 | 1.98 | 1.22 | 0.04 | 0.39 | 1.17 | 0.53 | 4.5 |
| PARA-Drive (Weng et al., 2024) | Det&Track&Map&Motion&Occ | 0.40 | 0.77 | 1.31 | 0.83 | 0.07 | 0.25 | 0.60 | 0.30 | 5.0 |
| GenAD (Zheng et al., 2024b) | Det & Map & Motion | 0.36 | 0.83 | 1.55 | 0.91 | 0.06 | 0.23 | 1.00 | 0.43 | 6.7 |
| UAD-Tiny (Guo et al., 2024) | Det | 0.47 | 0.99 | 1.71 | 1.06 | 0.08 | 0.39 | 0.90 | 0.46 | 18.9† |
| UAD∗ (Guo et al., 2024) | Det | 0.39 | 0.81 | 1.50 | 0.90 | 0.01 | 0.12 | 0.43 | 0.19 | 7.2† |
| **SSR (Ours)** | None | **0.24** | **0.65** | **1.36** | **0.75** | **0.00** | **0.10** | **0.36** | **0.15** | **19.6** |
| ST-P3‡ (Hu et al., 2022) | Det & Map & Depth | 1.33 | 2.11 | 2.90 | 2.11 | 0.23 | 0.62 | 1.27 | 0.71 | 1.6 |
| UniAD∗‡ (Hu et al., 2023) | Det&Track&Map&Motion&Occ | 0.44 | 0.67 | 0.96 | 0.69 | 0.04 | 0.08 | 0.23 | 0.12 | 1.8† |
| VAD-Tiny‡ (Jiang et al., 2023) | Det & Map & Motion | 0.46 | 0.76 | 1.12 | 0.78 | 0.21 | 0.35 | 0.58 | 0.38 | 16.8 |
| VAD-Base‡ (Jiang et al., 2023) | Det & Map & Motion | 0.41 | 0.70 | 1.05 | 0.72 | 0.07 | 0.17 | 0.41 | 0.22 | 4.5 |
| BEV-Planner‡ (Li et al., 2024c) | None | 0.28 | 0.42 | 0.68 | 0.46 | 0.04 | 0.37 | 1.07 | 0.49 | - |
| PARA-Drive‡ (Weng et al., 2024) | Det&Track&Map&Motion&Occ | 0.25 | 0.46 | 0.74 | 0.48 | 0.14 | 0.23 | 0.39 | 0.25 | 5.0 |
| LAW‡ (Li et al., 2024b) | None | 0.26 | 0.57 | 1.01 | 0.61 | 0.14 | 0.21 | 0.54 | 0.30 | 19.5 |
| GenAD‡ (Zheng et al., 2024b) | Det & Map & Motion | 0.28 | 0.49 | 0.78 | 0.52 | 0.08 | 0.14 | 0.34 | 0.19 | 6.7 |
| SparseDrive‡ (Sun et al., 2024) | Det & Track & Map & Motion | 0.29 | 0.58 | 0.96 | 0.61 | 0.01 | 0.05 | 0.18 | 0.08 | 9.0 |
| UAD∗‡ (Guo et al., 2024) | Det | 0.28 | 0.41 | 0.65 | 0.45 | 0.01 | **0.03** | 0.14 | 0.06 | 7.2† |
| **SSR‡ (Ours)** | None | **0.18** | **0.36** | **0.63** | **0.39** | **0.01** | 0.04 | **0.12** | **0.06** | **19.6** |

## 4.2 IMPLEMENTATION DETAILS

**Settings** We build up SSR on VAD (Jiang et al., 2023) and follow the setting of VAD-Tiny. We adopt ResNet-50 (He et al., 2016) as image backbone operating at an image resolution of $640 \times 360$. The BEV representation is generated at a $100 \times 100$ resolution and then compressed into sparse scene tokens with shape $16 \times 256$. The number of navigation commands remains 3 as prior works (Hu et al., 2023; Jiang et al., 2023). Other settings follow VAD-Tiny unless otherwise specified. In closed-loop simulation, we utilize ResNet-34 (He et al., 2016) as the image backbone, resizing the input image size to $900 \times 256$. The target point is concatenated with driving commands as the navigation information. The TCP head (Wu et al., 2022) is applied for planning module.

**Training Parameters** Our open-loop model is trained for 12 epochs on 8 NVIDIA RTX 3090 GPUs with a batch size of 1 per GPU. The training phase costs about 11 hours which is $13\times$ faster than UniAD. We utilize the AdamW (Loshchilov & Hutter, 2019) optimizer with a learning rate set to $5\times10^{-5}$. The weight of imitation loss and BEV loss is both 1.0. The closed-loop model is trained for 60 epochs on 4 NVIDIA RTX 3090 GPUs with a batch size of 32 per GPU. The learning rate is set to $1\times10^{-4}$ while being halved after 30 epochs.

## 4.3 MAIN RESULT

**Open-Loop Evaluation** Our method outperforms existing E2EAD approaches in nuScenes, achieving superior results in both L2 error and collision rate, as shown in Tab. 1. For the well-known method UniAD, which employs most auxiliary tasks, our method reduces 0.28m (27.2% relatively) average $L2_{MAX}$ error and 0.16% (51.6% relatively) average $CR_{MAX}$ without any auxiliary tasks. When compared to our baseline method VAD-Tiny, SSR not only reduces 0.39m (50.0% relatively) average $L2_{AVG}$ error and 0.46% (79.3% relatively) average $CR_{AVG}$, but also outperforms the VAD-Base with an obvious margin (45.8% average $L2_{AVG}$ and 70.7% average $CR_{AVG}$ reducation relatively). Furthermore, our method demonstrates real-time efficiency, achieving 19.6 FPS (the latency analysis in Appendix D), which is $10.9\times$ faster than UniAD and $4.3\times$ faster than VAD-Base. Remarkably, it is also $2.2\times$ faster than the prior sparse work SparseDrive (Sun et al., 2024), while reducing the average $L2_{AVG}$ error by 0.22m.

Table 2: **Performance on Town05 Long benchmark.**

| Method | Modality | DS↑ | RC↑ | IS↑ |
|---|---|---|---|---|
| CILRS (Codevilla et al., 2019) | C | 7.8 | 10.3 | 0.75 |
| LBC (Chen et al., 2020) | C | 12.3 | 31.9 | 0.66 |
| Transfuser (Prakash et al., 2021) | C+L | 31.0 | 47.5 | 0.77 |
| Roach (Zhang et al., 2021) | C | 41.6 | **96.4** | 0.43 |
| ST-P3 (Hu et al., 2022) | C | 11.5 | 83.2 | - |
| TCP (Wu et al., 2022) | C | 57.2 | 80.4 | 0.73 |
| VAD-Base (Jiang et al., 2023) | C | 30.3 | 75.2 | - |
| ThinkTwice (Jia et al., 2023b) | C+L | 65.0 | 95.5 | 0.69 |
| DriveAdapter(Jia et al., 2023a) | C+L | 65.9 | 94.4 | 0.72 |
| **SSR (Ours)** | C | **78.9** | 95.5 | **0.83** |

Table 3: **Component-wise Ablation.**

| Modules | | L2 (m) ↓ | | | | CR (%) ↓ | | | |
|---|---|---|---|---|---|---|---|---|---|
| STL | FFP | 1s | 2s | 3s | Avg. | 1s | 2s | 3s | Avg. |
| | | 0.23 | 0.65 | 1.41 | 0.76 | 0.04 | 0.58 | 0.66 | 0.43 |
| ✓ | | **0.23** | **0.64** | 1.39 | 0.75 | 0.02 | 0.10 | 0.47 | 0.20 |
| ✓ | ✓ | 0.24 | 0.65 | **1.36** | **0.75** | **0.00** | **0.10** | **0.36** | **0.15** |

Table 4: **Number of Scene Queries.**

| Number | L2 (m) ↓ | | | | CR (%) ↓ | | | |
|---|---|---|---|---|---|---|---|---|
| | 1s | 2s | 3s | Avg. | 1s | 2s | 3s | Avg. |
| 8 | **0.22** | **0.59** | **1.25** | **0.69** | 0.04 | 0.14 | 0.43 | 0.20 |
| 16 | 0.24 | 0.65 | 1.36 | 0.75 | **0.00** | **0.10** | 0.36 | **0.15** |
| 32 | 0.26 | 0.67 | 1.38 | 0.77 | 0.04 | 0.12 | **0.31** | 0.16 |
| 64 | 0.30 | 0.74 | 1.47 | 0.84 | 0.18 | 0.39 | 0.66 | 0.41 |

Table 5: **Ablation of navigation guidance.** GS means *go straight* and LR denotes *turn left / right*.

| Navigation Guidance | L2-GS (m) ↓ | | | | L2-LR (m) ↓ | | | | CR-GS (%) ↓ | | | | CR-LR (%) ↓ | | | |
|---|---|---|---|---|---|---|---|---|---|---|---|---|---|---|---|---|
| | 1s | 2s | 3s | Avg. | 1s | 2s | 3s | Avg. | 1s | 2s | 3s | Avg. | 1s | 2s | 3s | Avg. |
| ✗ | 0.24 | 0.61 | 1.31 | 0.72 | 0.34 | 0.96 | 1.98 | 1.09 | 0.09 | 0.38 | 0.40 | 0.29 | 0.00 | 0.44 | 1.90 | 0.78 |
| ✓ | **0.23** | **0.61** | **1.28** | **0.71** | **0.33** | **0.91** | **1.88** | **1.04** | **0.00** | **0.08** | **0.18** | **0.10** | **0.00** | **0.29** | **1.70** | **0.66** |

When compared to previous approaches that eliminate auxiliary annotations, SSR demonstrates impressive performance across all metrics. LAW (Li et al., 2024b), for instance, achieves a similar inference speed to SSR but retains a substantial gap in both L2 error and collision rate. The method closest in performance to ours is UAD (Guo et al., 2024), using a larger ResNet-101 backbone and a $1600 \times 900$ image resolution input, and requires an additional open-set 2D detector to supervise objectness information. Despite these additional resources, UAD still shows a 0.15m higher average $L2_{MAX}$ error compared to SSR, along with a 2.7× lower inference speed.

**Closed-Loop Evaluation** As presented in Tab. 2, our method significantly outperforms existing works in terms of driving score, including those utilizing LiDAR input (Jia et al., 2023b;a). For camera-based methods, SSR achieves a 31.7-point improvement in driving score over TCP and a remarkable 2.6× increase over VAD-Base. SSR also achieves the highest infraction score, demonstrating its comprehensive capabilities and robust performance in challenging environments.

## 4.4 ABLATION STUDY

### 4.4.1 COMPONENT-WISE ABLATION

In Tab. 3, we present an ablation study on the proposed components. When the STL is enabled instead of directly interacting waypoint queries with the BEV feature, the collision rate is reduced by more than half. This significant decrease underscores the STL's ability to effectively distill critical information from the dense scene data, thereby minimizing the impact of irrelevant features and reducing computational redundancy. Furthermore, when incorporating the Future Feature Predictor, we observe a further reduction in the average collision rate to 0.15%. This improvement highlights the Future Feature Predictor's role in enhancing SSR's comprehension of scene dynamics, contributing to more safe trajectory planning and overall performance gains.

### 4.4.2 NUMBER OF SCENE QUERIES

We evaluate the impact of varying the number of scene queries in Tab. 4. For L2 error, using 8 queries yields the best performance, with performance declining as the number of queries increases. Interestingly, when considering the collision rate, the optimal performance is achieved with 16 queries. Therefore, we select 16 queries as the default setting in SSR to strike a balance between minimizing L2 error and reducing collision rate. The poor performance observed with 64 scene queries suggests that an excessive number of queries may overwhelm the model with too much perception information, leading to confusion similar to directly interacting with dense BEV features.

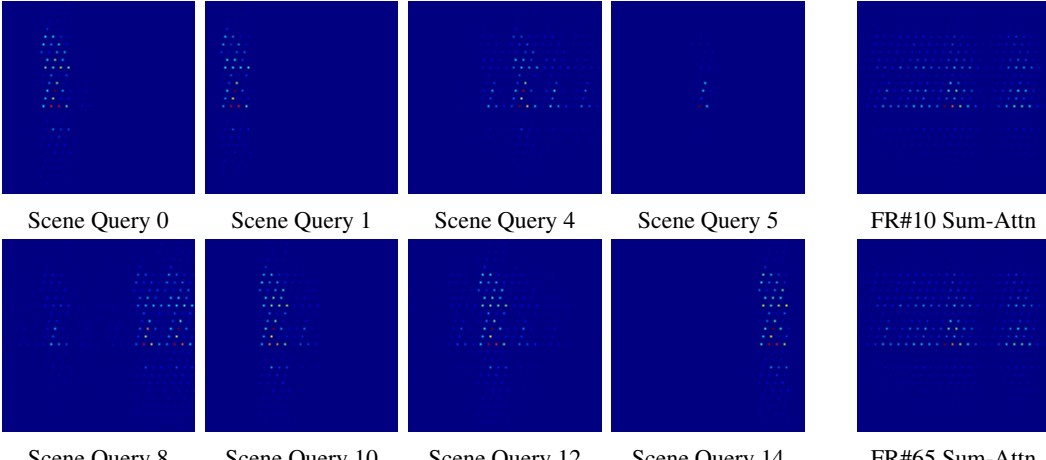

| Scene Query 0 | Scene Query 1 | Scene Query 4 | Scene Query 5 | FR#10 Sum-Attn |
|---|---|---|---|---|

| Scene Query 8 | Scene Query 10 | Scene Query 12 | Scene Query 14 | FR#65 Sum-Attn |
|---|---|---|---|---|

Figure 5: **Visualization of BEV Square Attention Map of Scene Queries.** Attention maps for 8 of the 16 tokens are displayed. Ego vehicle is located at the center while up direction indicates the front of ego. Brighter areas represent higher attention weights. The full set is provided in Appendix A.

Figure 6: **Sum-Attention Map in Different Scenes.** FR: frame number.

## 4.5 Analysis and Discussion

**How does scene queries represent the scene?** To understand why only a handful of queries can effectively represent the entire scene and even outperform more complex designs, we visualize 8 out of the 16 BEV square attention maps $\varpi(\mathbf{B}_t^{navi})$ from the STL module in Fig. 5. The results reveal that each query focuses uniformly on a distinct region of the BEV space, with different queries attending to different areas. When summing the attention maps of all scene queries, we observe that the sum-attention map surprisingly covers the entire scene in a balanced manner, with greater emphasis on the front region than the back. Furthermore, the attention maps remain relatively consistent across different frames, as shown in Fig. 6, indicating that $\varpi(\mathbf{B}_t^{navi})$ offers stable spatial compression guidance for SSR. In essence, the scene queries act as a compressed representation of dense BEV features, where each query concentrates on a specific spatial region.

**What does scene queries learn?** In Fig. 7, we visualize the navigation-aware BEV features $\mathbf{B}_t^{navi}$ as background color map and highlight the most positive activation positions of the scene queries across different scenarios. When overtaking a vehicle on the left as shown in Fig. 7(a), the activation positions primarily focus on the overtaken vehicle and the left rear area, anticipating potential risks. In a straightforward driving scenario (Fig. 7(b)), the scene queries are more dispersed, with attention directed towards a front-right vehicle, potentially anticipating a cut-in. During a right turn at an intersection (Fig. 7(c)), the scene queries not only activate around the right rear vehicle but also pay attention to the left crosswalk, where pedestrians might appear. When we further compare the same command in different scenes between Fig. 7(b) and Fig. 8(b), the scene queries focus on different regions adaptively. These observations demonstrate how our sparse scene queries can understand various scenes and manage different driving scenarios.

**How does navigation information work?** In Tab. 5, We conduct experiments to assess the effect of navigation commands in different scenarios, demonstrating that navigation guidance improves planning results across all cases. We further test different commands within the STL module for the same frame at an intersection, and visualize the corresponding scene queries in Fig. 8. Compared to the original command *go straight*, when the command is changed to *turn left*, the module shifts its attention to pedestrians on the left. Similarly, with the *turn right* command, the STL module increases its focus on the front-right area, particularly highlighting a vehicle on the right that is not prominent in the other navigation scenarios. These findings demonstrate that navigation commands effectively guide the STL module to extract relevant information from the scene. Additionally, we conduct experiments to evaluate the impact of confusing navigation commands in Appendix E.

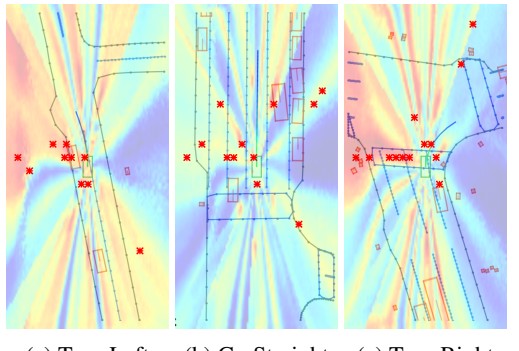 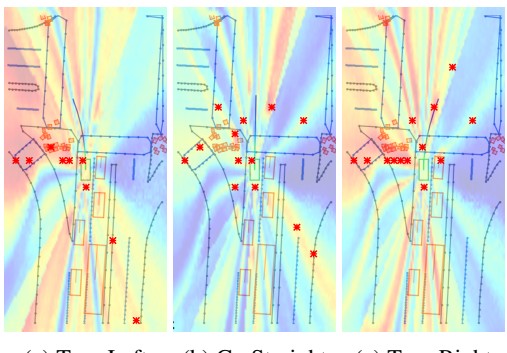

(a) Turn Left    (b) Go Straight    (c) Turn Right       (a) Turn Left    (b) Go Straight    (c) Turn Right

Figure 7: **Visualization of Scenes Queries in Different Scenarios.** The central green box represents ego vehicle. The red boxes indicate the ground truth object, while the dotted lines denote ground truth map. The red star marker is the most activated position of each scene query.

Figure 8: **Visualization of Scene Queries for Different Navigation Commands in Same Scene.** Different navigation commands are input to the SSR within the same scene to investigate their impact on scene queries. The original command is *go straight*.

To summarize above discussions, we revisit the generation of scene query in Eq. 5, where $\mathbf{B}_t^{navi}$ encodes the navigation information, and $\varpi_i(\mathbf{B}_t^{navi})$ is responsible for spatial compression:

$$\mathbf{s}_i = \rho(\ \underbrace{\mathbf{B}_t^{navi}}_{NaviGuidance}\ \odot\ \underbrace{\varpi_i(\mathbf{B}_t^{navi})}_{SpatialCompression}\ ). \qquad (16)$$

### 4.6 VISUALIZATION

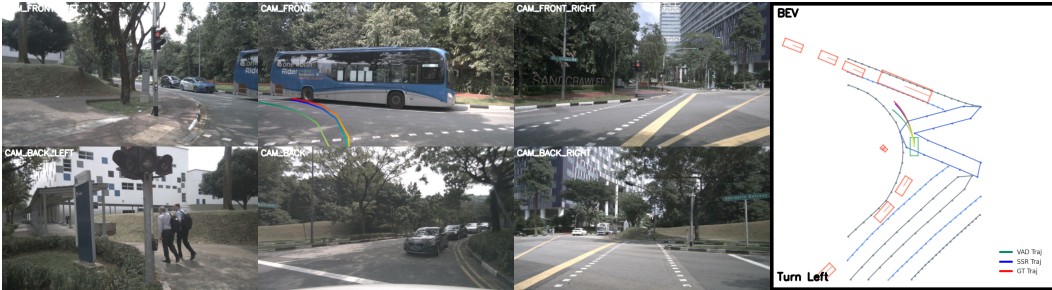

Figure 9: **Visualization of Planning Results.** The perception results are rendered from annotations.

In Fig. 9, we present a qualitative result of SSR on planning trajectories, demonstrating strong alignment with the ground truth compared to VAD-Base. Additional visualizations across various scenes, including failure cases, can be found in the Appendix B and C due to space limitations.

## 5 CONCLUSION

The SSR framework presents a significant advancement in the field of E2EAD by challenging the conventional reliance on perception tasks. By utilizing navigation-guided sparse tokens and temporal self-supervision, SSR addresses the limitations of perception-heavy architectures, achieving state-of-the-art performance with minimal costs. Moreover, visualization of the sparse tokens enhances the interpretability and transparency of SSR's navigation-guided process. We hope SSR can provide a strong foundation for scalable, interpretable, and efficient autonomous driving systems.

**Limitations and Future Work.** Despite these advances, current simple navigation commands may constrain SSR's effectiveness in more complex driving scenarios. Future work will explore integrating more sophisticated navigation prompts, such as routing and natural language.

ACKNOWLEDGMENTS

This research is supported by National Key R&D Program of China (2022YFB4300300).

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

## A    VISUALIZATION FOR ALL ATTENTION MAP OF SCENE QUERIES

We visualize the BEV square attention maps of all 16 scene queries used in Fig. 10. It can be observed that each scene query captures information from different spatial regions with a uniform attention distribution. When these queries are combined to represent the entire BEV feature, it effectively compresses the spatial space while applying different attention weights.

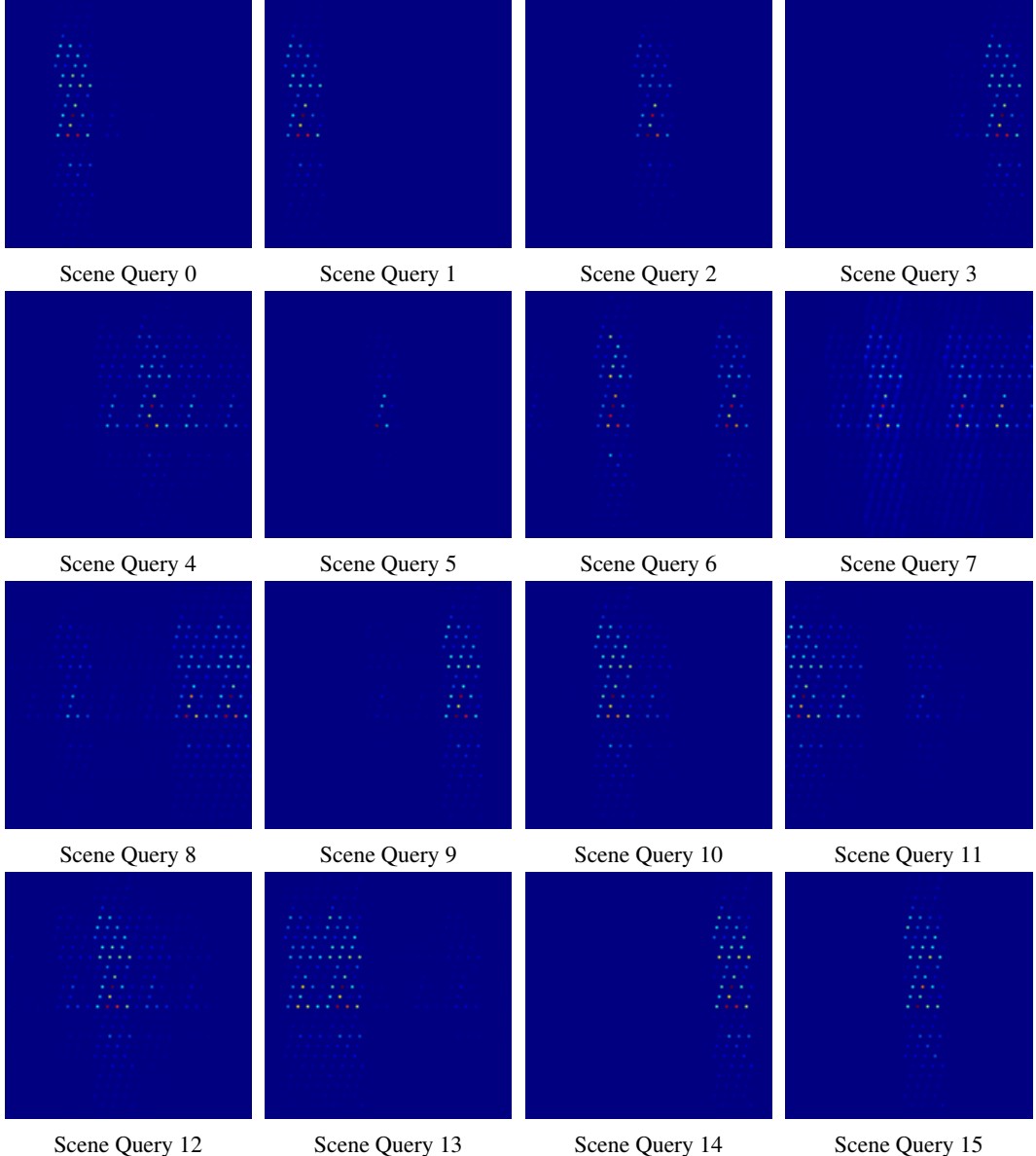

Figure 10: **Visualization of All BEV Square Attention Map of Scene Queries in Same Scene.** The ego vehicle is located at the center, with the upward direction indicating the front of the vehicle. Brighter areas represent higher attention weight.

## B    QUALITATIVE RESULTS IN DIFFERENT SCENES

As illustrated in Fig. 11, we visualize additional planning trajectories of SSR across various scenes. In Fig. 11(a), SSR achieves an even smoother result than the ground truth when turning left. When

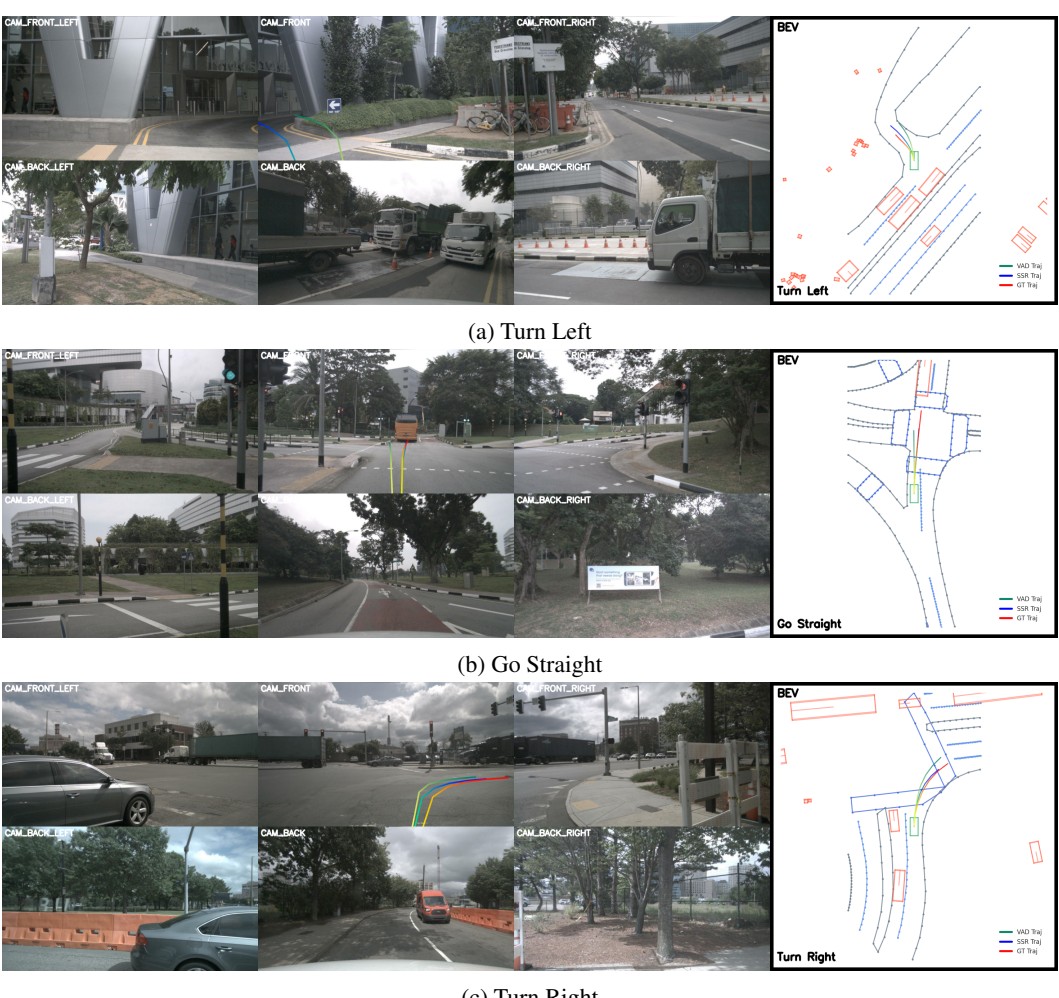

(a) Turn Left

(b) Go Straight

(c) Turn Right

Figure 11: **Visualization of Planning Trajectories in Different Scenes.** The perception information is rendered from annotations. The dashed lines denote the map of the scene, while red boxes represent for the object detection. Ego vehicle is drawn by the green box in center.

following a bus through an intersection with the *go straight* command, as shown in Fig. 11(b), our method also outperforms the baseline method, VAD, by producing a better trajectory. Similarly, in the *turn right* scenario illustrated in Fig. 11(c) , SSR demonstrates superior performance compared to VAD, achieving a smaller turning radius. These results demonstrate that SSR effectively understands the scene and produces accurate planning results through imitation learning. Even without explicit perception supervision, the learned scene queries capture the essential elements of complex scenes for autonomous driving.

## C   FAILURE CASES

We identified some failure cases for SSR, which are visualized in Fig. 12, highlighting the two most common reasons. The first common issue is noise in the initial frame of a clip for temporal module, which is also discussed in Weng et al. (2024). Due to zero initialization of input features and the ego vehicle's state, the temporal module of E2EAD methods can be negatively impacted, leading to inferior performance, as illustrated in Fig. 12(a). The second issue arises from ambiguous navigation commands in the label generation of nuScenes dataset. For example in Fig. 12(b), the command is to *go straight*, but the ground truth trajectory involves turning left to change lanes. Since SSR heavily relies on navigation commands, it can be adversely affected when encountering ambiguous

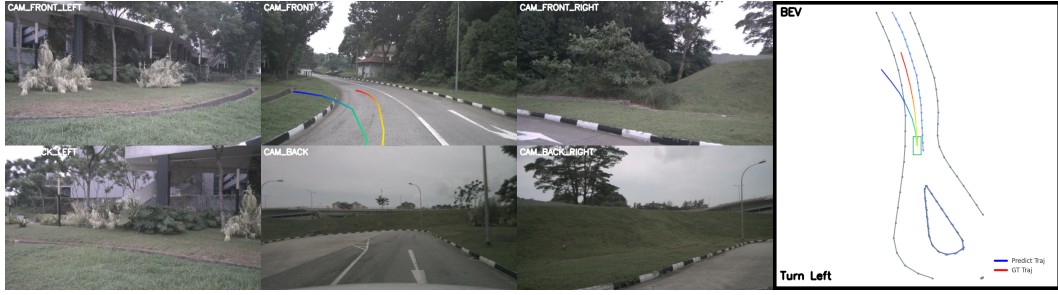

(a) Failure Reason: Noisy First Frame for Temporal Module

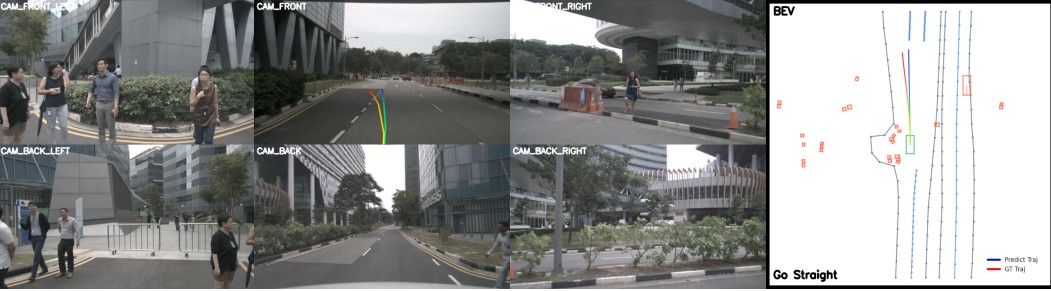

(b) Failure Reason: Ambiguous Navigation Commands in the Dataset

Figure 12: **Visualization of Common Failure Cases.**

instructions. However, these reasons which heavily related to dataset can be easily avoided in real world scenarios by integrating true navigation command and run in real-time image sequences.

# D    LATENCY ANALYSIS

Inference latency analysis of SSR components is presented in Fig. 13. The evaluation was conducted on an NVIDIA GeForce RTX 3090 GPU with a batch size of 1. The image backbone and encoder, responsible for generating dense BEV features, contribute to 90.7% of the total latency. In contrast, our proposed Scenes TokenLearner incurs only 7.8% of the latency, highlighting its efficiency in extracting useful information from massive dense BEV feature. The planning decoder, which interacts way point queries with the scene queries and output final planning trajectory, adds just 1.5% to the latency, as SSR only utilizes 16 tokens to represent the scenes.

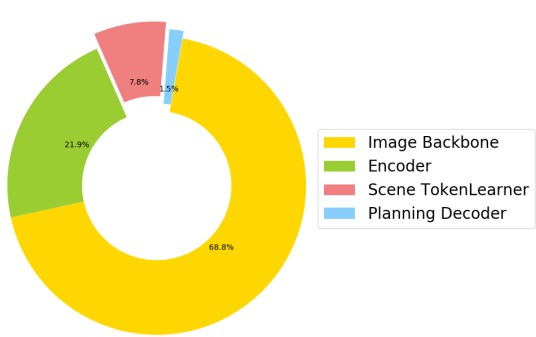

Figure 13: **Latency Analysis.**

# E    DISCUSSIONS

**Can SSR learn sufficient perception information for AD Task?** Although our framework eliminates the need for perception annotations, we investigate the effect of incorporating perception branches in Tab. 6. We follow approach in PARA-Drive (Weng et al., 2024) to integrate auxiliary tasks into SSR, using them to supervise the BEV feature in parallel with the planning task. In addition, to better evaluate the framework in ablation studies, we measure Curb Collision Rate (CCR) as introduced in Li et al. (2024c). The results indicate that even without a supervised object detection module, our method achieves a lower collision rate. Similarly, the CCR is even lower when SSR operates without a map branch. These findings suggest that SSR can effectively learn the necessary

Table 6: **Perception tasks ablation.** CCR calculated in 0.1m resolution following Li et al. (2024c).

| Auxiliary Task | | L2 (m) ↓ | | | | CR (%) ↓ | | | | CCR (%) ↓ | | | |
|---|---|---|---|---|---|---|---|---|---|---|---|---|---|
| Map | Obs | 1s | 2s | 3s | Avg. | 1s | 2s | 3s | Avg. | 1s | 2s | 3s | Avg. |
| | | **0.24** | **0.65** | **1.36** | **0.75** | **0.00** | **0.10** | **0.36** | **0.15** | **0.19** | 1.01 | **2.71** | **1.30** |
| ✓ | | 0.29 | 0.71 | 1.44 | 0.81 | 0.13 | 0.16 | 0.68 | 0.32 | 0.20 | **0.90** | 2.85 | 1.32 |
| | ✓ | 0.30 | 0.70 | 1.37 | 0.79 | 0.06 | 0.31 | 0.68 | 0.35 | 0.25 | 1.25 | 3.40 | 1.63 |
| ✓ | ✓ | 0.33 | 0.77 | 1.50 | 0.86 | 0.02 | 0.16 | 0.59 | 0.26 | 0.21 | 1.13 | 2.87 | 1.40 |

perception information for autonomous driving tasks without explicit supervision, maintaining high performance across all metrics.

However, we follow the 12-epoch training setup for fairness, which may lead to suboptimal perception performance. To investigate further, we conduct additional experiments with SSR trained using pretrained weights from 48 epochs of perception learning, as employed in VAD (Jiang et al., 2023). The pretrained model achieves 27.99 mAP and 40.15 NDS for obstacles and 48.78 mAP for mapping on nuScenes. As illustrated in Tab. 7, the pretrained weights does not significantly enhance trajectory prediction performance. This suggests that SSR inherently learns scene understanding in a different way than traditional perception-tasks-driven approaches.

Table 7: **Effect of Pretrained Perception Modules.**

| Pretrain | L2 (m) ↓ | | | | CR (%) ↓ | | | | CCR (%) ↓ | | | |
|---|---|---|---|---|---|---|---|---|---|---|---|---|
| | 1s | 2s | 3s | Avg. | 1s | 2s | 3s | Avg. | 1s | 2s | 3s | Avg. |
| × | **0.24** | **0.65** | **1.36** | **0.75** | **0.00** | **0.10** | **0.36** | **0.15** | **0.19** | **1.01** | **2.71** | **1.30** |
| ✓ | 0.48 | 1.03 | 1.82 | 1.11 | 0.25 | 0.31 | 1.42 | 0.66 | 0.67 | 1.61 | 4.02 | 2.10 |

**Impact of confusing driving commands.** Since E2EAD models typically lack HD maps, a high-level driving command input is necessary for navigation. To evaluate the effect of confusing commands, we test four scenarios: consistently "go straight," consistently "turn left," consistently "turn right," and random commands in Tab. 8. Notably, as the ego vehicle typically operates on the left side of the road, "turn left" and "go straight" commands lead to comparable L2 error and collision rates. However, the "turn right" command shows an obvious increase in collision rate, often resulting in conflicts with oncoming vehicles. Random commands cause a noticeable degradation in performance but still produce reasonable results, demonstrating the model's resilience to noisy navigation inputs. These findings highlight the strengths of our approach while identifying opportunities to improve its handling of ambiguous or conflicting commands.

Table 8: **Effect of Driving Commands.**

| Command | L2 (m) ↓ | | | | CR (%) ↓ | | | |
|---|---|---|---|---|---|---|---|---|
| | 1s | 2s | 3s | Avg. | 1s | 2s | 3s | Avg. |
| Original | **0.24** | **0.65** | **1.36** | **0.75** | **0.00** | **0.10** | **0.36** | **0.15** |
| Go Straight | 0.25 | 0.67 | 1.40 | 0.77 | 0.04 | 0.16 | 0.51 | 0.23 |
| Turn Left | 0.26 | 0.68 | 1.40 | 0.78 | 0.00 | 0.12 | 0.55 | 0.22 |
| Turn Right | 0.26 | 0.70 | 1.44 | 0.80 | 0.10 | 0.23 | 1.27 | 0.53 |
| Random | 0.26 | 0.68 | 1.41 | 0.78 | 0.06 | 0.14 | 0.72 | 0.31 |

