# OpenReview forum: "Navigation-Guided Sparse Scene Representation for End-to-End Autonomous Driving"
_ICLR.cc/2025/Conference — ICLR 2025 Poster_

### Official Review · Reviewer_4p2A · 2024-10-24

**Soundness:** 3
**Presentation:** 3
**Contribution:** 3
**Rating:** 6
**Confidence:** 5

**Summary:**

This work proposes to extract sparse scene queries to represent the scene and use the scene queries to generate planning results.  Further, they propose an auxiliary task head and loss - predict future BEV feature.

**Strengths:**

1. The elimination of middle task could indeed reduce the latency and annotation requirements.
2. The writing is clear

**Weaknesses:**

1. Lack of closed-loop evaluation:  as pointed in AD-MLP[1] and BEVPlanner[2], it is rather simple to fit nuScenes, as most of its data is going straight.  BEVPlanner only uses BEV feature, even without the need of the proposed scene queries.  (**Strangely, the authors cite [2] but do not compare with [2] in Table 1. I hope the authors could give an explanation**). Not to mention that AD-MLP even does not need any sensor inputs.  Thus,  **it is not convincing to claim that there is no need for middle supervision under nuScenes open-loop evaluation**. It is important to have closed-loop results and ablation studies on it.

[1] Rethinking the Open-Loop Evaluation of End-to-End Autonomous Driving in nuScenes.

[2] Is Ego Status All You Need for Open-Loop End-to-End Autonomous Driving? CVPR 2024

Due to the missing of closed-loop results, the argument in the paper is not convincing and thus I give a reject rating. I will increase my rating if closed-loop results and ablations are provided.

**Questions:**

None

---

> ### Author Response · Authors · 2024-11-18
> **Thanks and Response to Reviewer 4p2A (1/2)**
>
> Thanks for your valuable review which helps us improve this work. We address your concerns below.
>
> ### **Closed-Loop Experiments**
>
> We appreciate the reviewer's emphasis on closed-loop evaluation. We conduct closed-loop experiments using the CARLA simulator \[1\], leveraging the widely adopted Town05 Long benchmark to evaluate performance. The training dataset consists of 189K frames collected by Roach \[2\] at 2 Hz across 4 CARLA towns (Town01, Town03, Town04, and Town06), following previous works \[3-5\]. The training data has no overlap with Town05 Long benchmark.
>
> We utilized the official CARLA metrics for evaluation:
>
> - Route Completion (RC): Percentage of the route completed.
> - Infraction Score (IS): Measures infractions, including collisions with pedestrians, vehicles, road layout, and traffic signals.
> - Driving Score (DS): Main metric, calculated as the product of RC and IS.
>
> We utilize ResNet-34 as the image backbone, resizing the input image size to 900 $\times$ 256. For STL module, we concatenate the command, target point and current speed to a MLP as navigation information. The TCP head \[3\] is applied for planning module. Below are our closed-loop results, which will be added to the revised paper detailedly. **We also attach the video clips in supplementary material.**
>
> | Method | Modality | DS $\uparrow$ | RC $\uparrow$ | IS $\uparrow$ |
> | --- | :---: | :---: | :---: | :---: |
> | CILRS | Camera | 7.8 | 10.3 | 0.75 |
> | LBC | Camera | 12.3 | 31.9 | 0.66 |
> | Transfuser | Camera&LiDAR | 31.0 | 47.5 | 0.77 |
> | Roach | Camera | 41.6 | **96.4** | 0.43 |
> | ST-P3 | Camera | 11.5 | 83.2 | \-  |
> | LAV | Camera&LiDAR | 46.5 | 69.8 | 0.73 |
> | TCP | Camera | 57.2 | 80.4 | 0.73 |
> | VAD | Camera | 30.3 | 75.2 | \-  |
> | ThinkTwice | Camera&LiDAR | 65.0 | 95.5 | 0.69 |
> | DriveAdapter | Camera&LiDAR | 65.9 | 94.4 | 0.72 |
> | **SSR** | Camera | **78.9** | 95.5 | **0.83** |
>
> Our method significantly outperforms existing works in terms of DS, including those utilizing LiDAR input \[4, 5\]. For camera-based methods, SSR achieves a 31.7-point improvement in DS over TCP and a remarkable 2.6$\times$ increase over VAD. The closed-loop experiments demonstrate that our method performs reliably in complex, long-term driving scenarios, reinforcing the viability of eliminating middle supervision without compromising on safety or accuracy.
>
> To further investigate the impact of our proposed modules, we perform an ablation study in the closed-loop setting. The results below demonstrate that both modules contribute significantly to performance. The STL module yields substantial improvements across all metrics while BWM module further enhances the driving score and infraction score.
>
> | STL | BWM | DS $\uparrow$ | RC $\uparrow$ | IS $\uparrow$ |
> | --- | --- | :---: | :---: | :---: |
> |     |     | 62.6 | 90.3 | 0.72 |
> | ✔   |     | 72.7 | **97.4** | 0.75 |
> | ✔   | ✔   | **78.9** | 95.5 | **0.83** |
>
> ### **Comparison with BEV-Planner**
>
> We believe the comparison with BEV-Planner \[6\] was ignored in the initial review. In fact, we **did include BEV-Planner results in Tab. 1** (Line 343). The comparison of Tab. 1 is based on two different protocols:
>
> - The upper part compares methods using the UniAD protocol.
> - The lower part compares methods using the VAD protocol.
>
> This distinction is explained in Lines 314-321. BEV-Planner appears only in the lower section because it reports results using the VAD protocol in its paper. Due to the lack of released code, we could not include UniAD protocol results for BEV-Planner. We ensured a fair and consistent comparison by maintaining protocol alignment within each section of the table.
>
> We hope this clarifies the situation. The omission of BEV-Planner from the upper part of the table is due to protocol differences and not an oversight on our part.

---

> ### Author Response · Authors · 2024-11-18
> **Thanks and Response to Reviewer 4p2A (2/2)**
>
> ### **Evaluation on nuScenes**
>
> We noted the significant impact of using ego status in the planning module on L2/CR metrics revealed in \[6\], so we did not incorporate it in our experiments to ensure a robust design. Additionally, BEV-Planner introduced two solutions for more adequate nuScenes evaluation:
>
> - The introduction of Curb Collision Rate (CCR) metric, which methods relying on ego status may struggle.
> - The distinction between go-straight (GS) and turn left/right (LR) scenarios when calculating L2 and CR metrics, which provides a finer-grained assessment.
>
> These solutions are both applied in our experiments in Tab.3 and Tab. 4. In response to the reviewer’s concerns on the open-loop nuScenes results, we provide more CCR comparisons and scenario-based results in VAD protocol below:
>
> | Method/CCR (%) | 1s $\downarrow$ | 2s $\downarrow$ | 3s $\downarrow$ | Avg $\downarrow$ |
> | --- | :---: | :---: | :---: | :---: |
> | UniAD | 0.21 | 1.32 | 3.63 | 1.72 |
> | VAD-Base | 0.60 | 2.38 | 5.18 | 2.72 |
> | GoStraight | 2.07 | 8.09 | 15.7 | 8.62 |
> | Ego-MLP | 0.27 | 2.52 | 6.60 | 2.93 |
> | BEV-Planner | 0.70 | 3.77 | 8.15 | 4.21 |
> | **SSR** | **0.15** | **0.46** | **1.31** | **0.64** |
>
> | Method | Avg L2-GS (m) $\downarrow$ | Avg L2-LR (m) $\downarrow$ | Avg CR-GS (%) $\downarrow$ | Avg CR-LR (%) $\downarrow$ |
> | --- | :---: | :---: | :---: | :---: |
> | BEV-Planner | 0.48 | 0.81 | 0.40 | 2.25 |
> | SSR | **0.37** | **0.53** | **0.04** | **0.19** |
>
> These comparisons demonstrate that SSR is effective in both straightforward and complex scenarios on nuScenes, supporting its adaptability beyond simple cases.
>
> We appreciate the opportunity to discuss the role of open-loop evaluation. While we recognize the limitations of open-loop testing, we believe that dismissing it entirely overlooks valuable insights. The crux lies in ensuring an appropriate distribution of testing data and relevant metrics, as also suggested by recent research in simulator design, such as NAVSIM \[7\]. In this spirit, we believe our results on the nuScenes dataset, alongside our visualizations, provide meaningful insights into our model’s design and functionality.
>
> \[1\] CARLA: an open urban driving simulator. CoRL, 2017
>
> \[2\] End-to-end urban driving by imitating a reinforcement learning coach. ICCV, 2021
>
> \[3\] Trajectory-guided control prediction for end-to-end autonomous driving: a simple yet strong baseline. NeurIPS, 2022
>
> \[4\] Think twice before driving: towards scalable decoders for end-to-end autonomous driving. CVPR, 2023
>
> \[5\] Driveadapter: Breaking the coupling barrier of perception and planning in end-to-end autonomous driving. ICCV, 2023
>
> \[6\] Is Ego Status All You Need for Open-Loop End-to-End Autonomous Driving? CVPR 2024
>
> \[7\] NAVSIM: Data-Driven Non-Reactive Autonomous Vehicle Simulation and Benchmarking. NeurIPS, 2024

---

> > ### Comment · Reviewer_4p2A · 2024-11-18
> >
> > The experiments of closed-loop experiments and ablation study solve my concerns. The scores on CARLA is impressive. I increase my score to 6. The insights in this paper are meaningful to the community.
> >
> > The closed-loop study should be highlighted in main text and hopefully the closed-loop code could be released as the score is very strong.

---

> > > ### Author Response · Authors · 2024-11-19
> > > **Thanks for the feedback**
> > >
> > > Thank you again for your dedicated review and constructive feedback. We are pleased to have addressed your concerns and will incorporate your suggestions to further improve our manuscript.
> > >
> > > Additionally, we are actively working on refactoring and releasing our code to ensure transparency and facilitate future research.

---

### Official Review · Reviewer_zc5J · 2024-10-28

**Soundness:** 3
**Presentation:** 2
**Contribution:** 3
**Rating:** 6
**Confidence:** 3

**Summary:**

The paper introduces a novel SSR framework that requires no auxiliary task outputs or labels, relying instead on just 16 sparse navigation tokens for guidance. This approach achieves SOTA performance on the nuScenes dataset, with both inference and training processes being fast.

**Strengths:**

The model proposed in this paper yields excellent results, achieving SOTA performance on the nuScenes dataset. Additionally, both the inference and training speeds are fast on 3090 GPUs.

**Weaknesses:**

1. From the ablation studies (Table 2), it can be seen that the core contribution of this paper is the introduction of navigation information as guidance. The addition of the STL and BWM modules does not significantly affect L2, which suggests that navigation information provides significant assistance for lane following, especially when driving straight. It is understandable that the introduction of the BWM module, which incorporates temporal modeling, improves collision rates. In other words, the key contribution of this paper lies in utilizing navigation information, which is not employed by other methods. This kind of approach is relatively common in the industry but is rarely mentioned in academia. Whether merely introducing navigation information as input is worthy of an ICLR paper is something I am somewhat conflicted about.
2. The STL and BWM modules lack novelty. Furthermore, I'm not particularly fond of naming the BWM module a "world model." Given that it primarily involves predicting the next frame in BEV, the term "world model" might be considered too broad. I'd like to reiterate that I think the core contribution of this paper is the introduction of navigation information.
3. The color scheme used in the figures could be slightly unappealing, with most modules and features using white as the primary color. It would be nicer if the visuals were more aesthetically pleasing.

**Questions:**

Refer to the weakness

---

> ### Author Response · Authors · 2024-11-18
> **Thanks and Response to Reviewer zc5J (1/2)**
>
> Thank you for your helpful comments. Following are our detailed responses to your concerns.
>
> ### **Clarification of Navigation Information**
>
> As the reviewer observed, navigation information is often overlooked in prior academic work, and to our knowledge, its use in industry is also primarily limited to the planning module. In the contrast, our method **leverages navigation information to directly guide the perception process**, rather than just serving as input for the planning module. This distinction is visually illustrated in Fig. 2 of our paper.
>
> We understand the reviewer’s observation that improvements in lane-following scenarios dominate the ablation results in Table 2. This is partially due to dataset bias in nuScenes, where lane-keeping scenarios are prevalent. To address this, we additionally conduct closed-loop evaluations using the CARLA simulator on the widely adopted Town05 Long benchmark.
>
> For evaluation, we utilized official CARLA metrics:
>
> - Route Completion (RC): Percentage of the route completed.
> - Infraction Score (IS): Measures infractions, including collisions with pedestrians, vehicles, road layout, and traffic signals.
> - Driving Score (DS): Main metric, calculated as the product of RC and IS.
>
> The results below demonstrate that both modules contribute significantly to performance. The STL module significantly improves all metrics, while the BWM module further enhances DS and IS:
>
> | STL | BWM | **DS** $\uparrow$ | RC $\uparrow$ | IS $\uparrow$ |
> | --- | --- | :---: | :---: | :---: |
> |     |     | 62.6 | 90.3 | 0.72 |
> | ✔   |     | 72.7 | **97.4** | 0.75 |
> | ✔   | ✔   | **78.9** | 95.5 | **0.83** |
>
> Furthermore, we want to clarify that the effect of navigation guidance is evaluated in Tab. 3 instead of Tab.2 of our paper. The experiments in Tab. 3 demonstrate that navigation guidance improves planning results across all scenarios, rather than only lane following scenarios.
>
> ### **Clarification of BWM Module**
>
> Despite the advances of STL module, our navigation-guided perception design introduces unique challenges in temporal modeling. As noted in Lines 284-286, scene queries may focus on different elements in consecutive frames. Prior query-level pairwise temporal modeling approaches \[1, 2\] fail to address this mismatch, causing non-convergence in our architecture.
>
> To overcome this limitation, our novel BWM module leverages BEV-level self-supervision to stabilize features across frames, achieving a further improvement in collision rate (Tab. 2) and outperforming prior methods (Tab. 1). Additionally, the component-wise ablation in closed-loop experiments emphasizes the importance of the BWM module, as shown in the table above.
>
> We understand the reviewer’s concern regarding the term “world model.” Our "world model" is applied to BEV features across consecutive frames, which compensates the absence of perception task supervision. We recognize that this term might appear too broad, as the module is primarily focused on predicting the next frame in BEV rather than building a full world model. If this naming is confusing, we are open to renaming it to something more descriptive, such as **"BEV Prediction Module"** or **"Future Feature Predictor"**, to better reflect its function in predicting the future state of the environment.

---

> ### Author Response · Authors · 2024-11-18
> **Thanks and Response to Reviewer zc5J (2/2)**
>
> ### **Contribution and Novelty of SSR**
>
> The design of the STL and BWM modules is grounded in leveraging the **planning-focused nature of E2E frameworks.** Specifically, inspired by human drivers' attention mechanisms, we propose that E2EAD systems can selectively focus on relevant elements, rather than extracting all scene elements exhaustively in previous works.
>
> The primary contribution of our paper lies in establishing a human-inspired framework for E2EAD, significantly reduces computational costs by adaptively focusing on essential parts of scenes. This shift on paradigm emphasizes planning-centric design over exhaustive perception, presenting an more efficient and practical alternative to traditional architectures.
>
> Within this framework, the STL module implements the selective perception strategy, while the BWM module enhances temporal coherence across frames. These two modules are integral and complementary components of our novel navigation-guided perception design of SSR. Their contributions should therefore be assessed in the context of the overall framework rather than in isolation.
>
> We also provide the results in closed-loop experiments below, while attaching the video clips in supplementary material.
>
> | Method | DS $\uparrow$ | RC $\uparrow$ | IS $\uparrow$ |
> | --- | :---: | :---: | :---: |
> | CILRS | 7.8 | 10.3 | 0.75 |
> | LBC | 12.3 | 31.9 | 0.66 |
> | Transfuser | 31.0 | 47.5 | 0.77 |
> | Roach | 41.6 | **96.4** | 0.43 |
> | ST-P3 | 11.5 | 83.2 | \-  |
> | LAV | 46.5 | 69.8 | 0.73 |
> | TCP | 57.2 | 80.4 | 0.73 |
> | VAD | 30.3 | 75.2 | \-  |
> | ThinkTwice | 65.0 | 95.5 | 0.69 |
> | DriveAdapter | 65.9 | 94.4 | 0.72 |
> | **SSR** | **78.9** | 95.5 | **0.83** |
>
> The superior performance of SSR in both open-loop and closed-loop experiments demonstrates its capability to explore deeper into the potential of E2EAD systems. This novel framework, which integrates navigation-guided selective perception and temporal coherence, goes beyond the simple introduction of navigation information. It sets a new benchmark for practical and efficient autonomous driving solutions.
>
> ### **Color Scheme**
>
> Our figures utilize a white background for traditional elements and a colored background to highlight the unique contributions of our modules. We appreciate the reviewer’s suggestion regarding aesthetics and will incorporate improvements in the revised version to enhance clarity and appeal. For instance:
>
> - In Fig. 2, the white elements will be replaced with a more visually distinct background to emphasize critical features.
> - In Fig. 4, module colors will be adjusted to correspond to their respective blocks, using a cohesive color scheme (e.g., red hues for STL and blue hues for BWM).
>
> \[1\] Enhancing end-to-end autonomous driving with latent world model. arXiv
>
> \[2\] End-to-end autonomous driving without costly modularization and 3d manual annotation. arXiv

---

> > ### Comment · Reviewer_zc5J · 2024-11-26
> >
> > Thanks to the authors for their detailed responses.
> >
> > 1. I acknowledge that the nuScenes dataset contains more cases of driving straight, which introduces bias. The results on CARLA are more convincing to me.
> > 2. Applying navigation information to the perception module rather than the planning module indeed has novelty. I apologize for my earlier misunderstanding.
> > 3. Regarding the naming of the BWM module, I still hold my original opinion that "world model" is too broad. Predicting future BEV features for supervision has also been explored in previous works, such as HOP[1] and MIM4D[2], please including citations to these works.
> >
> > Based on the above points, I have decided to raise the score of this paper to 6.
> >
> > [1] Zong Z, Jiang D, Song G, et al. Temporal enhanced training of multi-view 3d object detector via historical object prediction[C]//Proceedings of the IEEE/CVF International Conference on Computer Vision. 2023: 3781-3790.
> >
> > [2] Zou J, Liao B, Zhang Q, et al. MIM4D: Masked Modeling with Multi-View Video for Autonomous Driving Representation Learning[J]. arXiv preprint arXiv:2403.08760, 2024.

---

> > > ### Author Response · Authors · 2024-11-26
> > > **Thanks for the feedback**
> > >
> > > We sincerely appreciate your thoughtful feedback and recognition of our contributions, including the novelty of applying navigation information to the perception module.
> > >
> > > We understand your concerns about the BWM module. We have renamed the module as "Future Feature Predictor" and revised the manuscript to include citations you mentioned to better contextualize our work. Thank you again for your time and for reconsidering our submission.

---

### Official Review · Reviewer_rfNp · 2024-10-30

**Soundness:** 3
**Presentation:** 3
**Contribution:** 3
**Rating:** 6
**Confidence:** 4

**Summary:**

In this paper, the authors propose a navigation-guided sparse scene representation for end-to-end autonomous driving. The proposed solution eliminates the need for supervised sub-tasks, allowing computational resources to concentrate on essential elements directly related to navigation intent. Specifically, the proposed solution utilizes learned sparse query representations guided by navigation commands, and introduces a BEV world model for self-supervision on dynamic scene changes to highlight the critical role of temporal context in autonomous driving. The experimental results show that the proposed solution achieves great performance on nuSccenes dataset with minimal training and inference cost.

**Strengths:**

1. The proposed solution sounds solid in theory. Specifically, eliminating the need for supervised sub-tasks, e.g., detection, mapping, motion, occupancy, etc., indeed speed up the training and inference time. The proposed navigation-guided concept and BEV world model should extract useful features for the autonomous driving.
2. The proposed navigation-guided concept is inspring. As mentioned in the paper, it is a human-inspried solution whose intuition is to mimic the behavior of human beings. It is a good research direction. The authors propose a way to combine the information with the special features.
3. The ablation study is helpful. Readers can get more information from the ablation study results.
4. The analysis and discussion guide readers think more deeper about the model design and the performance.

**Weaknesses:**

1. There are some important information is missing. For example, detailed model structure, navigation commands pre-processing operation, etc.
2. It will be better if the authors could report some failure cases. Especially for those cases which the predictions are totally opposite toward the navigation command.

**Questions:**

1. Are the navigation commands open-set or close-set, how many commands were used in this paper, how to pre-processing the high-level navigation commands so that the model can take them as inputs, etc.
2. What is the performance of the model if the navigation command never appears in the training set (or confusing command, e.g., turn left and right, or turn left or right, etc.)?

---

> ### Author Response · Authors · 2024-11-18
> **Thanks and Response to Reviewer rfNp**
>
> Thank you for your thoughtful comments and constructive feedback. We address each of your points below:
>
> **Response to W1 & Q1:** Our model builds upon the VAD-Tiny [1] architecture. Specifically, we utilize ResNet-50 as the image backbone,  while the BEV encoder adopts the approach from BEVFormer [2]. To enhance navigation awareness, we modify the TokenLearner [3] into BEV space to extract scene queries. The planning decoder incorporates cross-attention, while the BWM module predominantly employs self-attention. We will release the code for our full model structure, providing further clarity on the implementation.
>
> In our experiments on the nuScenes dataset, we use three navigation commands (go straight, turn left, turn right). The preprocessing of these commands follows prior practice in VAD by transforming ground truth ego trajectories into driving commands based on the final future trajectory offset. This approach is also documented in the open-source VAD [code](https://github.com/hustvl/VAD/blob/main/tools/data_converter/vad_nuscenes_converter.py#L452).
>
> ```python
> # drive command according to final fut step offset from last context frame
> if ego_fut_trajs[-1][0] >= 2:
>     command = np.array([1, 0, 0])  # Turn Right
> elif ego_fut_trajs[-1][0] <= -2:
>     command = np.array([0, 1, 0])  # Turn Left
> else:
>     command = np.array([0, 0, 1])  # Go Straight
> ```
>
> **Response to W2:** We appreciate your suggestion to report failure cases. Our appendix (Line 749, Page 14) has provided visualizations and detailed analysis of two common failure types, one of which is related to ambiguous navigation commands.
>
> **Response to Q2:** Since end-to-end autonomous driving (E2EAD) models typically lack HD maps, a high-level driving command input is necessary for navigation. To address the reviewer's concerns about potentially confusing commands, we conducted additional experiments alongside our failure case analysis.
>
> | Command | Avg L2 (m) $\downarrow$ | Avg CR (%) $\downarrow$ |
> | --- | --- | --- |
> | Go Straight | 0.77 | 0.23 |
> | Turn Left | 0.78 | 0.22 |
> | Turn Right | 0.8 | 0.53 |
> | Random | 0.78 | 0.31 |
> | Original | **0.75** | **0.15** |
>
> For these experiments, we tested four additional command types: all go straight, all turn left, all turn right, and random commands. Notably, as the ego vehicle typically operates on the left side of the road, "turn left" and "go straight" commands led to comparable L2 error and collision rates. However, the "turn right" command showed an obvious increase in collision rate, often resulting in conflicts with oncoming vehicles. Random commands caused a noticeable degradation in performance but still produced reasonable results, demonstrating the model’s resilience to noisy navigation inputs. These findings highlight the strengths of our approach while identifying opportunities to improve its handling of ambiguous or conflicting commands.
>
> [1] Vad: Vectorized scene representation for efficient autonomous driving. In ICCV, 2023.
>
> [2] Bevformer: Learning bird’s-eye-view representation from multi-camera images via spatiotemporal transformers. In ECCV, 2022
>
> [3] Tokenlearner: Adaptive space-time tokenization for videos. In NeurIPS, 2021

---

> > ### Comment · Reviewer_rfNp · 2024-11-24
> >
> > Thanks to the authors for their detailed responses. The authors provided some details of the model structures and the results of confusing commands. These responses help address my concerns. Thanks.

---

> > > ### Author Response · Authors · 2024-11-25
> > > **Thanks for the feedback**
> > >
> > > We sincerely thank you for your thoughtful comments and the time you’ve invested in reviewing our manuscript. We are delighted that the additional details regarding the model structure and the results of confusing commands have addressed your concerns.
> > >
> > > Given the efforts we’ve undertaken to resolve the raised issues, would you be open to discussing a potential rate change? We would greatly appreciate your consideration. Again, thank you for your helpful comments! Your valuable input has greatly improved our manuscript.

---

### Official Review · Reviewer_pPL3 · 2024-11-03

**Soundness:** 3
**Presentation:** 3
**Contribution:** 3
**Rating:** 8
**Confidence:** 4

**Summary:**

This paper proposed a novel approach for trajectory prediction in autonomous driving called SSR. A very straight intuition is that the more comprehensive and precise the information one can extract, the better the downstream tasks will perform. Many recent models try to extract dense BEV features with rich scene information and route the information to multiple auxiliary tasks. On the contrary, the authors argue that for end-to-end autonomous driving, auxiliary supervision from perception tasks might be unnecessary, instead computational resources can be focused on crucial parts of the scene.
To achieve this, the paper introduced a scene token learner that learns sparse scene queries from BEV features extracted by a BEV encoder (BEVFormer in particular) and current navigation command. The trajectory is then predicted based on the scene queries. To train the model, two loss terms are used. The first term directly minimizes the L1 loss between the predicted and ground truth trajectory. The second term aims to enhance scene representation by introducing a BEV world model. This term replaces the perception sub-tasks in other models, with more focus on temporal context. The BEV world model takes the BEV features, scene queries and predicted trajectory as input and it tries to predict the BEV features at the next frame. We minimize the L2 distance between the output of the world model and the BEV features directly extracted by the BEV encoder at the next frame. The overall loss is the sum of these two terms.
The authors show that a small set of tokens is sufficient to outperform sota models in trajectory prediction on NuScenes dataset.

**Strengths:**

The paper is well written, with clear and insightful motivation. The proposed model architecture is novel, and the experimental results show significant improvement over existing methods. The experiment section is comprehensive, with insightful analysis.
In autonomous driving, multi-head models are very popular, for a number of reasons. Not only because different tasks can serve as auxiliary supervision for each other and contribute to the overall performance improvement, the output of the individual tasks can also be utilized by downstream tasks in a modular autonomous driving system (e.g. occupancy map can be the input of a traditional pathfinding algorithm for parking). However in a fully end-to-end autonomous driving system all those intermediate outputs are unnecessary. It’s interesting to see papers questioning the design choices of existing models and exploring alternative designs.

**Weaknesses:**

1. The proposed architecture borrows the idea of a world model by predicting future bev features. However it seems that the BEV features are only regularized by the trajectory prediction task. There’s no reconstruction loss and other downstream tasks. In this case it’s questionable to still call it a world model.
2. The paper lacks theoretical analysis of why the proposed SSR method is better.

**Questions:**

1. There’s a paper “Rethinking the Open-Loop Evaluation of End-to-End Autonomous Driving in nuScenes” (https://arxiv.org/abs/2305.10430) claims that a simple MLP model that takes simple ego state (velocity, acceleration, historical trajectory) can also outperform VAD and UniAD. While their claim is that the metrics may not adequately capture the superiority of different methods, in combination with table 2.(b) in this paper being reviewed, where a larger number of scene queries leads to worse performance, is this an indication of overfitting of the larger models (or generally speaking, a case where a larger model is being overperformed by a smaller model due to various reasons)? Considering the trajectories in NuScenes are relatively simple, will this weaken the conclusion of this paper, especially when the navigation tasks become more complex?
2. Why the visualization in Figure. 5 in a hexagon format instead of square grids?

---

> ### Author Response · Authors · 2024-11-18
> **Thanks and Response to Reviewer pPL3**
>
> Thank you for your valuable comments and constructive feedback. We address each of your points below:
>
> **Response to W1:** We appreciate your feedback regarding the term "world model" for the BWM module. We recognize that this term might appear too broad, as the module is primarily focused on predicting the next frame in BEV (Bird's Eye View) rather than building a full world model. If this naming is confusing, we are open to renaming it to something more descriptive, such as **"BEV Prediction Module"** or ****"Future Feature Predictor"****, to better reflect its function in predicting the future state of the environment.
>
> **Response to W2:** Thank you for suggesting theoretical analysis. Our work is primarily empirical; however, we aim to make SSR’s navigation-guided perception process more interpretable by visualizing scene queries. These visualizations reveal how SSR’s perception module prioritizes scene elements based on navigational input, distinguishing it from traditional BEV architectures.
>
> **Response to Q1:** We note the significant impact of using ego status in the planning module on L2/CR metrics revealed in \[1\] and \[2\], so we did not incorporate it in our experiments to ensure a robust design.
>
> Regarding the reviewer's concern about the nuScenes dataset, we conducted additional closed-loop experiments using the CARLA simulator, leveraging the widely adopted Town05 Long benchmark to evaluate performance. The training dataset consists of 189K frames collected by Roach \[3\] at 2 Hz across 4 CARLA towns (Town01, Town03, Town04, and Town06), following previous works \[4-6\]. The training data has no overlap with Town05 Long benchmark.
>
> We utilized the official CARLA metrics for evaluation:
>
> - Route Completion (RC): Percentage of the route completed.
> - Infraction Score (IS): Measures infractions, including collisions with pedestrians, vehicles, road layout, and traffic signals.
> - Driving Score (DS): Main metric, calculated as the product of RC and IS.
>
> We utilize ResNet-34 as the image backbone, resizing the input image size to 900 $\times$ 256. For STL module, we concatenate the command, target point and current speed to a MLP as navigation information. The TCP head \[4\] is applied for planning module. Below are our closed-loop results, which will be added to the revised paper. **We also attach the video clips in supplementary material.**
>
> | Method | Modality | DS $\uparrow$ | RC $\uparrow$ | IS $\uparrow$ |
> | --- | :---: | :---: | :---: | :---: |
> | CILRS | Camera | 7.8 | 10.3 | 0.75 |
> | LBC | Camera | 12.3 | 31.9 | 0.66 |
> | Transfuser | Camera&LiDAR | 31.0 | 47.5 | 0.77 |
> | Roach | Camera | 41.6 | **96.4** | 0.43 |
> | ST-P3 | Camera | 11.5 | 83.2 | \-  |
> | LAV | Camera&LiDAR | 46.5 | 69.8 | 0.73 |
> | TCP | Camera | 57.2 | 80.4 | 0.73 |
> | VAD | Camera | 30.3 | 75.2 | \-  |
> | ThinkTwice | Camera&LiDAR | 65.0 | 95.5 | 0.69 |
> | DriveAdapter | Camera&LiDAR | 65.9 | 94.4 | 0.72 |
> | **SSR** | Camera | **78.9** | 95.5 | **0.83** |
>
> Our method significantly outperforms existing works in terms of DS, including those utilizing LiDAR input \[5, 6\]. For camera-based methods, SSR achieves a 31.7-point improvement in DS over TCP and a remarkable 2.6$\times$ increase over VAD. These results indicate the comprehensive capabilities of SSR in long-term driving scenarios.
>
> We also appreciate your interest in extending SSR to more complex navigation commands, as noted in our limitations. We plan to explore such extensions in future work, potentially incorporating large language models (LLMs) to enable natural language-based navigation tasks.
>
> **Response to Q2:** The visualizations in Figure 5 are actually in square grids, not hexagonal format. The hexagonal appearance arises from the lower attention values in the grid corners, which may appear hexagonal due to the natural gradient in the attention map. We apologize for any confusion and will clarify this in our figure caption in revised version. For more comprehensive visualizations, please refer to Figure 10 (page 13) in the appendix, where we provide the full attention map set.
>
> \[1\] Rethinking the Open-Loop Evaluation of End-to-End Autonomous Driving in nuScenes. arXiv
>
> \[2\] Is Ego Status All You Need for Open-Loop End-to-End Autonomous Driving? CVPR 2024
>
> \[3\] End-to-end urban driving by imitating a reinforcement learning coach. In ICCV, 2021
>
> \[4\] Trajectory-guided control prediction for end-to-end autonomous driving: a simple yet strong baseline. NeurIPS, 2022
>
> \[5\] Think twice before driving: towards scalable decoders for end-to-end autonomous driving. In CVPR, 2023
>
> \[6\] Driveadapter: Breaking the coupling barrier of perception and planning in end-to-end autonomous driving. In ICCV, 2023

---

> > ### Comment · Reviewer_pPL3 · 2024-11-23
> >
> > Thanks to the authors for the detailed response. The close-loop evaluation addressed my concern about the nuScenes dataset. It's interesting to know that the grid corners have lower attention value. Since I already give accept rating I'll keep it unchanged.

---

> > > ### Author Response · Authors · 2024-11-24
> > > **Thanks for the feedback**
> > >
> > > Thank you again for your dedicated review and constructive feedback. We are glad that the additional experiments have addressed your concerns on nuScenes dataset.
> > >
> > > Again, thank you for your helpful comments! Your valuable input has greatly improved our manuscript.

---

### Official Review · Reviewer_odD2 · 2024-11-03

**Soundness:** 3
**Presentation:** 4
**Contribution:** 3
**Rating:** 6
**Confidence:** 3

**Summary:**

This paper presents a novel method for end-to-end trajectory prediction for autonomous driving, with a model that takes as input surround view camera images, and outputs a predicted trajectory for the vehicle. The main novelties are:
- the use of a sparse tokenizer to reduce the typical dense BEV representation into a sparse (in experiments 16) set of feature vectors.
- the introduction of the driving command into the BEV representation via cross attention.
- the application of a future prediction module for the BEV features as an auxiliary loss.
The proposed method aims to reduce the dimensionality of the BEV space without using pre-defined intermediate signals such as detections, mapping etc. This allows the model to be end-to-end differentiable, with the tokenization learned as a part of the driving task.

Experiments and ablations are performed on the nuScenes dataset, where the proposed method produces a new SOTA in terms of both quality and latency.

**Strengths:**

The main novel contribution of this work lies in the use of the TokenLearner for BEV feature compression. From the experiments, the authors demonstrate that they are able to achieve new SOTA trajectory prediction results without any intermediate perception signals used by previous works. Ablations are also performed demonstrating the improvements provided by both the future prediction auxiliary task as well as the navigation command cross attention into the BEV feature space. Visualizations are also provided demonstrating the the model's attention mechanism does seem to focus on reasonable parts of the scene given a navigation command (i.e. it's possible that the model is not directly overfitting in some way).

Overall the paper is well written and easy to understand, and the overall method seems reasonable and based on sound prior works.

**Weaknesses:**

The number of optimal scene queries selected by the authors seems surprisingly small, and may be skewed by the difficulty of the average scene encountered in nuScenes. From the provided visualizations, it looks like the model focuses directly on relevant agents and objects in the scene. However, it's unclear whether the model would be able to extend to extremely dense scenes, where a fixed number of scene queries may limit the models ability to handle many other agents / objects.

In Table 4, the authors ablate the addition of a perception module. However, no results for the perception tasks themselves are provided, and so it cannot be determined whether these tasks were actually trained appropriately (i.e. they may have simply diverged altogether).

One additional factor to call out is the inherent risk of the nuScenes dataset being skewed in some way (i.e. Rethinking the Open-Loop Evaluation of End-to-End Autonomous Driving in nuScenes), but this is not the fault of the authors.

In Figures 7 and 8, it would be helpful to explain the colormap.

**Questions:**

What is the channel dimension of the scene queries?

How would the model respond in a very dense scene? Is there an inherent limitation in the number of objects / components that the model can focus on given a fixed number of scene queries?

What is the quality of the perception outputs in Table 4?

---

> ### Author Response · Authors · 2024-11-18
> **Thanks and Response to Reviewer odD2**
>
> Thanks for your helpful comments. We address your concern below:
>
> **Response to W1 & Q2:** We appreciate the reviewer’s concerns regarding the number of scene queries. A key strength of our method lies in demonstrating that a minimal set of scene queries can effectively represent the entire scene without compromising performance. Our visualizations provide insight into how such a small number of queries can focus on critical traffic units and salient regions.
>
> Even in extremely dense scenes, we believe the number of interactive traffic units (e.g., key agents, critical road features) remains relatively small compared to the total scene complexity. This allows the model to handle dense environments without requiring a large number of queries. The results in Table 2(b) further support this, showing that increasing the number of scene queries beyond a certain point leads to suboptimal performance.
>
> However, we agree with the reviewer that adopting a dynamic number of scene queries could further enhance the model's flexibility and performance. A possible direction would be to start with a larger initial set of queries and predict a confidence score for each one. Queries falling below a predefined threshold could be discarded, enabling an adaptive number of queries tailored to the scene’s complexity. We leave this promising direction for further exploration.
>
> **Response to W2&Q3:** We appreciate the reviewer’s observation regarding the absence of perception metrics in Table 4. To address this, we provide the detailed results below:
>
> | Map | Obs | $mAP_{Map} \uparrow$ | $mAP_{Obs} \uparrow$ | $NDS_{Obs} \uparrow$ |
> | --- | --- | :---: | :---: | :---: |
> |     |     | \-  | \-  | \-  |
> | ✔   |     | 10.13 | \-  | \-  |
> |     | ✔   | \-  | 4.18 | 14.4 |
> | ✔   | ✔   | 10.72 | 3.96 | 12.67 |
>
> For fairness, we followed the 12-epoch training setup described in Line 360 of our revised paper. While this ensures consistency across experiments, it leads to lower perception performance compared to previous methods that train perception modules for 48 epochs before planning module's learning.
>
> To further address this concern, we conducted additional experiments by training SSR with pretrained weight from 48 epochs perception learning, as used in VAD. The pretrained model achieved 27.99 mAP and 40.15 NDS for obstacles and 48.78 mAP for mapping on nuScenes. However, even with these pretrained weights, SSR did not show significant improvements in trajectory prediction. This suggests that SSR inherently **learns scene understanding in a different way** than traditional perception-tasks-driven approaches.
>
> | Pretrain | Avg L2 (m) $\downarrow$ | Avg CR (%) $\downarrow$ | Avg CCR (%) $\downarrow$  |
> | --- | --- | :---: | :---: |
> | ✔   | 1.11 | 0.66 | 2.10 |
> | $\times$ | 0.75 | 0.15 | 1.30 |
>
> These results reinforce SSR’s distinct capability to bypass traditional perception modules, emphasizing its end-to-end nature and efficiency.
>
> **Response to W3:** To address the reviewer's concerns on nuScenes dataset, we additionally conducted closed-loop evaluations in the CARLA simulator, which demonstrate SSR’s robustness in diverse settings. We utilized the official CARLA metrics for evaluation:
>
> - Route Completion (RC): Percentage of the route completed.
> - Infraction Score (IS): Measures infractions, including collisions with pedestrians, vehicles, road layout, and traffic signals.
> - Driving Score (DS): Main metric, calculated as the product of RC and IS.
>
> Below results are evaluated on widely adopted Town05 Long benchmark. **We also attach the video clips in supplement materials.**
>
> | Method | Modality | DS $\uparrow$ | RC $\uparrow$ | IS $\uparrow$ |
> | --- | :---: | :---: | :---: | :---: |
> | CILRS | Camera | 7.8 | 10.3 | 0.75 |
> | LBC | Camera | 12.3 | 31.9 | 0.66 |
> | Transfuser | Camera&LiDAR | 31.0 | 47.5 | 0.77 |
> | Roach | Camera | 41.6 | **96.4** | 0.43 |
> | ST-P3 | Camera | 11.5 | 83.2 | \-  |
> | LAV | Camera&LiDAR | 46.5 | 69.8 | 0.73 |
> | TCP | Camera | 57.2 | 80.4 | 0.73 |
> | VAD | Camera | 30.3 | 75.2 | \-  |
> | ThinkTwice | Camera&LiDAR | 65.0 | 95.5 | 0.69 |
> | DriveAdapter | Camera&LiDAR | 65.9 | 94.4 | 0.72 |
> | **SSR** | Camera | **78.9** | 95.5 | **0.83** |
>
> The superior performance of SSR in both open-loop and closed-loop experiments demonstrates its capability to explore deeper into the potential of E2EAD systems.
>
> **Response to W4:** We appreciate the reviewer’s suggestion and add more detailed explanation of the colormap in Line 466 of revised paper. The colormap represents the **navigation-aware BEV features**, where warmer colors indicate higher attention weights.
>
> **Response to Q1:** The channel dimension of each scene query is **256**, following standard practice in previous works. We have added its dimension clearly in our revised paper.

---

> ### Comment · Reviewer_odD2 · 2024-11-23
>
> Thanks to the authors for their detailed responses. I believe that the additional experiments satisfy the questions and concerns about the paper.

---

> > ### Author Response · Authors · 2024-11-23
> > **Thanks for the feedback**
> >
> > We sincerely thank you for your kind acknowledgment and for appreciating our additional experiments and detailed responses. Your feedback has been invaluable in refining our work and highlighting its strengths.
> >
> > We are glad that the additional experiments have addressed your concerns regarding the paper. If you find our revisions and the newly added insights satisfactory, we would greatly appreciate your consideration of an updated score to reflect these improvements.
> >
> > Thank you again for your constructive guidance and support throughout this process!

---

### Author Response · Authors · 2024-11-22
**Revision Summary**

We sincerely thank the reviewers for their thoughtful feedback and constructive comments. Your insights have been invaluable in improving the quality of our manuscript.

We have carefully responded to each reviewer individually and revised our paper accordingly, with the changes clearly marked in blue. Below, we summarize the key updates:

- Incorporated closed-loop experiments and accompanying analysis into the main text. (Reviewers odD2, pPL3, zc5J, 4p2A)
- Renamed "BEV world model" to "Future Feature Predictor" and revised the corresponding sections. (Reviewers pPL3, zc5J)
- Expanded the explanations for visualizations to enhance clarity. (Reviewers odD2, pPL3)
- Revised the introduction to better articulate our contributions. (Reviewers zc5J, 4p2A)
- Added experiments and analysis on ambiguous commands in the appendix. (Reviewer rfNp)
- Conducted additional experiments on pretrained perception modules and moved the discussion to the appendix due to space limitations. (Reviewer odD2)
- Improved figure color schemes for better readability. (Reviewer zc5J)
- Included more citations to better contextualize our work.  (Reviewer zc5J)

We hope these revisions address all concerns and highlight the novelty, robustness, and practical implications of our approach. Thank you again for your valuable time and consideration.

---

### Meta-Review · Area_Chair_TMgD · 2024-12-23

**Metareview:**

The paper proposes navigation-guided sparse scene representation for end-to-end autonomous driving. The motivation is based on the expensive annotations when considering data scalability for real-time production. The core idea is designated for eliminating human-designed sub-tasks. SSR, the proposed method, achieved remarkable results in both performance and efficiency, surpassing previous SOTA (UniAD, e.g.)

There are five reviews for this paper. All the reviewers agree the paper is of sufficient novelty, solid experiments, well-written and easy to follow. Authors did a great job addressing reviewers concern (e.g. adding closed-loop experiments on Carla) and the manuscript is very clear to be ready for ICLR. AC agrees with the consensus.

**Additional Comments On Reviewer Discussion:**

One of the key concern is to add closed-loop experiments given recent advances mentioned by several reviewers. Authors have addressed the concern during rebuttal and answered reviewer's comment accordingly. As suggested by Reviewer and agreed by AC, it is strongly recommended to incorporate the closed-loop experiments, including all the revisions highlighed in blue, to the main text. Code should be released for the better advancement of this field.

---

### Decision · Program_Chairs · 2025-01-22

Accept (Poster)